***Nat Commun.* Author manuscript; available in PMC 2025 November 17.**

# A library of avian proteins improves palaeoproteomic taxonomic identification and reveals widespread intraspecies variability

**Maria C. Codlin**[1], **Lisa Yeomans**[2,3], **Josefin Stiller**[4], **Beatrice Demarchi**[1]

[1]Department of Life Sciences and Systems Biology, University of Turin, Turin, Italy.

[2]Globe Institute, Section for GeoBiology, University of Copenhagen, Copenhagen, Denmark.

[3]Institute of Archaeology, University College London, London, UK.

[4]Department of Biology, University of Copenhagen, Copenhagen, Denmark.

## Abstract

Biomineral-associated proteins, such as those found in bone, teeth, and egg-shell, have become instrumental for studying ancient life, as they often survive far longer than DNA. Harnessing advancements in avian genomics, we annotate bone and eggshell protein sequences for 112 Anatidae (ducks, geese and swans) species, a biologically complex group of birds that are central to many archaeological and ecological questions. While palaeoproteomics conventionally assumes that protein sequences vary only between-species or above, our research demonstrates widespread evidence for single amino acid polymorphisms (SAPs) occurring within-species, particularly within avian eggshell proteins. Furthermore, we construct a phylogenetic tree from 13 proteins that aligns with mtDNA-based phylogenies, while revealing highly variable topologies for individual protein trees, underscoring the need for caution when using fragmented proteins for taxonomic identification and determining evolutionary relationships. However, with comprehensive taxonomic coverage of Anatidae proteins, clear taxonomic patterns enable reliable

Correspondence to: Maria C. Codlin.

**Author contributions**
M.C.: conceptualization, methodology, validation, formal analysis, investigation, data curation, writing—original draft, writing—review and editing, visualisation, funding acquisition. L.Y.: conceptualization, writing—review and editing, visualisation, project administration, funding acquisition. J.S.: methodology, formal analysis, resources, writing—review and editing. B.D.: conceptualization, resources, writing—original draft, writing—review and editing, supervision, project administration, funding acquisition.

**Competing interests**
The authors declare no competing interests.

identification of bone and eggshell. We demonstrate this application to archaeological material from Teotihuacan, Mexico, and Shubayqa, Jordan. We highlight that extensive curated protein datasets accompanied by rigorous standards for assessing SAPs as taxonomic biomarkers are fundamental for correct taxonomic identification, setting benchmarks for palaeoproteomic applications in archaeology, ecology, and evolutionary biology.

---

Ancient biomolecules offer unprecedented insights into evolutionary pathways and are instrumental for improving the taxonomic identification of human, animal, plant and microbial remains recovered from archaeological and palaeoecological deposits, illuminating all facets of past cultural and natural environments[1–10]. Abundant or highly resistant biomineral-associated proteins, such as collagen type I in bone, amelogenin, ameloblastin and enamelin in dental enamel, or C-type lectins and ovocleidin in eggshell, often survive *post mortem* degradation[3,4,7,11–16]. When protein survival occurs, mass spectrometry techniques enable the reconstruction of ancient amino acid sequences that express mutations accumulated over time in coding regions of DNA. These amino acid polymorphisms form the basis for the identification of organisms and tissues, but a key question from evolutionary biologists and zooarchaeologists is the level of taxonomic resolution that these ancient proteins can offer.

Classic applications of palaeoproteomics (the study of ancient proteins by mass spectrometry) are based on collagen retrieved from osseous materials. Additionally, peptide fingerprinting applications such as "ZooMS - zooarchaeology by mass spectrometry" have found that, with some exceptions, bone collagen from mammals typically allows for taxonomic discrimination at the genus level[17]. While extensive research has continued to refine both the resolution and the confidence of collagen-based taxonomic identifications across multiple taxa[18–25], including species-level identifications of fish[21,25] and amphibians[23], the possibility of intraspecies diversity in protein sequences has not been systematically examined by the palaeoproteomics community. More recently, collagen and other protein sequences have been used to resolve the phylogenetic placement of human and animal fossils in the absence of preserved ancient DNA[5,6,8,26] albeit tree topologies based on fragments of ancient proteins may not correspond to those obtained by large-scale phylogenomics, particularly for more recent divergences, or may not be very robust[13,27].

Most of these applications are underpinned by the assumption that protein sequences do not vary within species but only at the level of species or above. Here we argue that close examination of this biological diversity, specifically single amino acid polymorphisms (SAPs), also offers glimpses of finer-grained variability both between and within species. Intraspecies SAPs have been identified in primates, specifically in ancient hominin and modern *Pongo* (orangutan) enamel and bone proteomes[28,29], including multiple apparent SAPs in recovered Denisovan collagen peptides[30]. If these patterns are confirmed in other systems, it could ultimately mean that evolutionary relationships may not be so easily reconstructed by using ancient amino acid sequences from a single specimen and that mixed phylogenetic signals may impact large-scale protein fingerprinting of ancient materials.

One of the major challenges in palaeoproteomic research thus far has been the limited availability of genomic data for a wide range of taxa, from which protein sequences may be

derived. We leverage opportunities offered by the rapid development of avian genomics[31–33] to annotate bird bone and eggshell sequences and systematically explore inter-taxon and intra-taxon variability in proteins employed in palaeoproteomic applications. We focus on Anatidae (ducks, geese and swans), birds which are often migratory, have specialised ecological requirements, frequently inter-breed, and have a long shared history with humans, making them biologically complex yet highly pertinent to archaeological and ecological questions. We present a phylogenetic tree based on these annotated proteins which is largely consistent with mtDNA based phylogenies. We also apply our protein dataset to precisely identify avian species at two sites, Tlajinga, Teotihuacan (Mexico) and Shubayqa (Jordan), which improves our understanding of human-avian relationships at key moments in human history. By targeting such a challenging taxonomic grouping, we provide theoretical and practical benchmarks for the application of palaeoproteomics in archaeology, ecology and evolutionary biology.

## Results

We annotated 1928 sequences from 160 Anatidae genomes, including 112 species from 43 genera for 13 proteins that are suitable for palaeoproteomic applications (Supplementary Data 1-3). These included type I collagen (COL1a1 and COL1a2) and selected eggshell proteins (c-type lectins XCA1 and XCA2, Ovocleidin116 (OC116), Albumin, BPI fold containing family B, member 4 (BPI-fold-B-4, BPIFB4), Clusterin, Lactadherin, Ovalbumin, Ovocalyxin32, Ovomucoid, and Ovotransferrin). A minimum coverage of 70% was used as a cut-off point for "complete" sequences used to analyse taxonomic variability (dataset 1, $n = 1832$), while both complete and partial sequences were included in a set of unique sequences used to derive markers for palaeoproteomic applications (dataset 2, $n = 1566$). Phylogenetic trees were created from each protein alignment in dataset 1 (Supplementary Figs. 2–14), as well as a concatenated alignment based on all 13 proteins (Fig. 1).

### Protein-based phylogenies of Anatidae

The concatenated phylogenetic tree (Fig. 1), covering 5553 amino acid positions, had high bootstrap resampling support (>90) for most branches, and was largely consistent with tree topologies expected from DNA data[34,35]. For individual protein trees, early branches corresponding to groups of species traditionally classified within the same tribe were well-supported (bootstrap resampling score >80) and these broad groupings were largely consistent with phylogenetic clades based on mtDNA[34–36]. For OC116, Albumin, Clusterin and Ovotransferrin, divergences for most genera were also well-supported, especially where multiple species were available for a given genus. For other proteins, divergences known to occur deeper in time, such as those between *Cygnus* (swans) and the *Anser*/*Branta* (geese) genera, and the early diverging *Dendrocygna* (whistling ducks)[34], were often well-supported, while more recent divergences were frequently poorly supported, even when species of the same genus appeared to cluster together. The topologies however varied remarkably by proteins, especially within the larger clade groupings, such as Anatini (dabbling ducks) (Fig. 2). In some proteins, such as OC116, *Mareca* and *Anas* were more closely related to each other than either is to *Spatula*, while in COL1a2, *Spatula* and *Anas* were indistinguishable, and *Mareca* was a distinct group. In XCA1 and XCA2, *Mareca* and

*Spatula* were more closely related to each other than to *Anas*. Therefore, while taxonomic information is undoubtedly stored in protein sequences, single-protein trees are not always able to track evolutionary relationships. This was further complicated by our observation that intraspecies variation (SAPs) occurred in annotated amino acid sequences in all protein types.

## Intra- and interspecies variation

In order to assess how variability in amino acid sequences differed across proteins, and therefore to examine their utility for palaeoproteomic applications, we analysed the pairwise intra- and interspecies distances for each protein (Fig. 3). This analysis was inspired by DNA barcoding gap studies[37–39], aiming to assess where variability was occurring between taxa at a broad level.

We found at least one instance of intraspecies variation (SAPs) in every protein, most commonly in BPIfoldB4 (13 of 27 species for which multiple individuals were available for comparison, henceforth denoted by "species[mi]"), Clusterin (9 of 14 species[mi]), OC116 (20 of 26 species[mi]) and Ovotransferrin (13 of 26 species[mi]), and seldom in Albumin (4 of 27 species[mi]), COL1a1 (2 of 11 species[mi]), and COL1a2 (3 of 26 species[mi]) (Fig. 3). Mean number of intraspecific SAPs (calculated across all genomes) was highest in Clusterin (2.8 SAPs per genome) and OC116 (6.2 SAPs per genome) and lowest in Albumin, COL1a1 and COL1a2 each. Mean SAPs may seem negligible for some proteins, including collagen and albumin (Fig. 3c), but Fig. 3b shows that even for these proteins, intraspecies SAPs were observed for 11-18% of species[mi], hence intraspecies variation is certainly not a rare occurrence and cannot be dismissed. Similarly, intraspecies distances were highest in OC116 and Clusterin, with mean SAPs per genome 12.8 and 4.8, respectively, and lowest in COL1a1 (0.1 SAPs), COL1a2 and Ovomucoid (both 0.2 SAPs) (Supplementary Data 4).

Overall, all eggshell proteins, except perhaps Ovomucoid, were more variable than the bone collagen proteins. These results demonstrate that, with large enough sequence datasets containing sufficient species[mi], variability in protein sequence becomes strikingly evident. Indeed, given that most species are still represented by a single genome, for many observed interspecies SAPs, we cannot distinguish which might be SAPs indicative of taxonomic differences from those that might reflect variation within a species.

For practical applications, e.g., in zooarchaeology or palaeontology, genera are useful categories for classifying ancient organisms into groups that share morphological and behavioural characteristics. Therefore, we investigated the frequency of SAPs in the most variable protein (OC116) across the traditional taxonomic units (Fig. 4). Many of the intergenus distance plots displayed a bimodal distribution, likely reflecting early evolutionary separation of Anatinae (most ducks) and Anserinae (geese and swans)[34]. Overall, we found that there was little to no overlap between the intergenus and interspecies distances for most genera, indicating that a large number of SAPs are shared among species of the same genus. Moreover, we identified intraspecies SAPs in most genera where pairwise comparisons were available, and *Mareca, Anas* and *Anser* were the most variable.

## Examining sources of intraspecies diversity

Given the unexpected finding of frequent intraspecies diversity in these protein sequences, we considered multiple sources of error, as well as biological processes that could result in SAPs (see methods). We first considered the annotation process as a potential source of spurious SAPs, yet our curated sequences are highly consistent with those available for RefSeq genomes (Supplementary Note 1, Supplementary Data 6)[40]. Next, the impact of sequencing and assembly errors was examined systematically for OC116, which had the highest observed variability and is frequently recovered from archaeological samples. We aligned all available sequence read archives (SRAs, $n = 8$) from seven *Anas platyrhynchos* (mallard) individuals (Supplementary Note 2) to the *OC116* gene. We observed that at least 10 or more reads were mapped to most single nucleotide polymorphisms (SNPs), suggesting assembly errors were not the cause of high variability in the genomes and supporting the validity of the observed SAPs.

We next performed variant calling to identify which SNPs were well supported for *OC116* and examine the frequency of variants across the *Anas platyrhynchos* sample population. Twenty-six SNPs were identified in protein coding regions of the gene, 16 of which are responsible for 15 missense mutations, i.e., leading to an SAP in the protein sequence, while nine were silent (Fig. 5). Four missense mutations were observed only in a single haploid chromosome, while 11 occurred ~50% of the time. All mutations observed in the *Anas platyrhynchos* OC116 protein alignment for which SRAs were available (Supplementary Fig. 15) were supported by variant calling, with one exception (see Supplementary Note 2), although that variant was found in two *Anas platyrhynchos* samples without SRAs available for evaluation. In three out of seven individuals, heterozygosity was observed for multiple variants.

Because multiple individuals were observed to be heterozygous at variant sites, we tested for the possibility of gene duplication by examining whether reads mapping to the *OC116* gene also mapped to other areas of the genome but found only a small number of reads that did so (0-6%, Supplementary Note 2). This indicates that observed intraspecies variability in OC116 was more likely the result of biological processes.

To rule out the possibility that intraspecies variation was observed in Anatidae due to the unique characteristics of this family, such as widespread hybridization and history of domestication, we annotated OC116 and Ovocalyxin-32 sequences from seven other species[mi] ($n = 32$, Supplementary Data 7–8). We observed intraspecies variability in OC116 in five species across five of the six orders of birds represented, including *Pelecanus crispus* (Dalmatian pelican), *Lycocorax pyrrhopterus obiensis* (Obi paradise-crow), *Accipiter gentilis* (Northern goshawk), *Rhynochetos jubatus* (kagu), and *Chlamydotis macqueenii* (MacQueen's bustard). We observed intraspecies variability in Ovocalyxin-32 in *Grus americana* (whooping crane), *Chlamydotis macqueenii* and *Lycocorax pyrrhopterus obiensis*.

The presence of intraspecies variation in collagen was especially surprising given the overall higher conservation of this protein (Fig. 3). We used the only available short-read dataset to assess the validity of the intraspecies variation in *Anser indicus* (bar-headed goose,

GCA025583725) from a specimen with a single intraspecies COL1a2 SAP at position 246. In this case, following alignment of reads to the COL1a2 reference gene for *Anser cygnoides* (swan goose), the SAP was found to be poorly supported, as only a single read matched this location after standard filtering. This demonstrates that while intraspecies variability was identified in all proteins, and the variability in OC116 was well supported, a small number of SAPs may result from sequencing or assembly errors.

## Discussion

From our analysis of 1832 the annotated sequences from 112 species of Anatidae, representing ~65% of all Anatidae species[41], we found unexpectedly high variation in some proteins, including intraspecies variation across all 13 protein types. The frequency of SAPs differed by protein type, as well as by taxon considered (i.e., taxonomic rank). As expected, the highly structured collagen proteins had fewer amino acid substitutions relative to length, while XCA1, OC116, Clusterin and Ovocalyxin-32 were the most diverse relative to length. Although intraspecies and interspecies variation was infrequent in many proteins, this study demonstrates that when a sufficient number of samples are analysed, the observation of SAPs occurring between and within species is common.

For OC116 we identified 15 missense mutations within *Anas platyrhynchos* alone and determined that heterozygosity was present in three out of seven individuals examined. DNA damage causes G-T or C-A nucleotide mutations, which is often associated with low-frequency variants (~1-5%)[42]. But none of those observed in *Anas platyrhynchos* resulted from these substitutions, although the one example of intraspecies variability we were able to examine for COL1a2 was the result of a poorly supported C-A SNP. Mutagenic damage, like other sequencing and assembly errors, is an infrequent but persistent issue in published genomic databases[33,42,43]. While our results demonstrate that most of the observed intraspecies protein variation is not an artefact of sequencing or assembly, but is occurring at the biological level, the evidence for variants caused by mutagenic damage emphasises the need for careful consideration of the SAPs chosen for taxonomic discrimination. Given that some, if rare, intraspecies variation does occur within collagen sequences of birds, as has been observed in orangutans and hominins[28,30], the application of collagen peptide mass fingerprinting (ZooMS) to identify archaeological samples should also consider this possibility when identifying biomarkers for taxonomic determination, particularly if the aim is to distinguish closely related groups. This is especially important for fish[21,25] and amphibians[23] for which interspecific variability is common. Crucially, our observation of both intra- and interspecies variation in all annotated protein groups complicates the standard assumption that SAPs observed between two species or genera can be used for taxonomic identification or evolutionary placement.

The observed intraspecies variation may have several biological causes, among which frequent hybridization among Anatidae[44,45] and a millennary history of human modification and management of Anatidae populations[46,47], which continues today at industrialised scales[48,49], certainly play a fundamental role. Variants in the genes involved in egg-laying and reproduction in domesticated chicken and duck breeds are known to produce changes in eggshell or reproductive traits, such as egg shape and weight, and clutch size, and are often

encouraged by agricultural specialists today[50–57]. Importantly, however, our observation of intraspecies variability across multiple orders of birds, including paradise-crow, goshawk, bustard, pelican, whooping crane and kagu, highlights the widespread nature of intraspecies protein variation among birds. Moreover, while most of the intraspecies variation we observed occurs in eggshell proteins associated with reproduction, many of these sequences have homologues in other animals, including ovocalyxin-32 and OC116[58]. Avian ovocleidin 116 is homologous to the matrix extracellular phosphoglycoprotein (MEPE) in mammals that plays a role in biomineralisation of bone[59]. As of 18/04/2025, there are at least 721 observed variants for Human MEPE on UniProt[60] indicating that this variation is not related specifically to avian reproduction. If intraspecies variation affects all organisms, then the answer to the question "what level of taxonomic resolution can ancient proteins offer?" needs to be carefully evaluated.

Evolutionary palaeoproteomics is perhaps the most exciting avenue for reconstructing the history of life, as it is able to leverage the genetic information encoded in ancient samples in which DNA is not preserved. We confirm that proteins describe divergences occurring in deep time, e.g., between large orders of birds such as Galliformes (landfowl), Anseriformes (waterfowl) and Columbiformes (pigeons and doves)[13,19,61,62], all between 70 and 60 mya[32], and families like Phasianidae (pheasants, partridges) and Odontophoridae (New World quails)[13,19,63], which separated 37 mya[32]. Between-species relationships for Anatidae have not yet been resolved by genome-wide studies. Multiple phylogenies are available based on mitochondrial genes or mitogenomes, which differ in topology[34,35,44,64]. Here we compare our protein-based phylogenies to Sun et al.[34], whose mitochondrial gene-based tree contains the most comparable range of species to our study. Our concatenated protein-based phylogeny of 13 nuclear-encoded genes demonstrates high bootstrap support (>90) for separating all genera (except *Chenonetta*). This includes recent divergences, such as *Lophonetta* + *Tachyeres* (5.3 mya), and *Amazonetta* + *Speculanas* (3.9 mya)[34]. In general, the relationships between genera are consistent with those presented by Sun et al.[34] with two major differences. Our protein phylogeny places *Cairina* and *Aix* in a separate clade rather than as a sister branch to *Tadorna* (diverging 12.5 mya) and *Chloephaga*. Similarly, our tree places *Oxyura* outside of Anserinae, inconsistent with the mitochondrial DNA tree (diverging 22.5 mya). Tree topologies vary more between the protein and mitochondrial DNA trees within genera, and species divergence is less well-supported by bootstrap scores, especially where speciation occurred within the last 2–3 million years (as estimated by Sun et al.[34]).

The congruence between the mtDNA and protein trees highlights the utility of protein sequences for encoding taxonomic information. However, our tree is based on multiple proteins covering 5553 amino acid positions, an amount of data that is unlikely to be generated by most palaeoproteomic studies. Moreover, for practical applications of palaeoproteomics it is important to note that proteins are tissue specific (i.e., collagen is recovered from bone and not eggshell), and degraded. Therefore, the utility of proteins for taxonomic identification changes according to substrate type and extent of molecular preservation. Nonetheless, multiple lines of evidence, such as morphology and species distributions, can be integrated to narrow down taxonomic identifications, and through rigorous assessment of phylogenetically informative SAPs in bone collagen and eggshell

proteins, palaeoproteomic applications can significantly improve our ability to identify avian species from fragmentary remains. We demonstrate this point by applying these reference sequences (dataset 2) to reevaluate ancient protein data from two previous studies that failed to identify archaeological avian remains below the family or genus level.

The data from two archaeological sites, *Tlajinga, Teotihuacan* (Mexico) and *Shubayqa* (Jordan), now give us a fine-grained picture of complex human-environment relationships in the past. Codlin et al.[19] published a large-scale application of ZooMS to avifaunal bone from Teotihuacan city, Mexico (200 BCE – 600 CE). They revealed a high diversity of aquatic birds consumed in the city, including four groups of archaeological specimens that were only able to be identified to family level (Anatidae). We reevaluated the four specimens representative of these groups, for which both LC-MS/MS and MALDI-TOF MS data was available, and used dataset 2 to refine identifications, relying on SAPs that are observed consistently across multiple individuals from the taxonomic groups concerned: widgeon (*Mareca* sp.), pintail (*Anas acuta*), ruddy duck (*Oxyura jamaicensis*) and a large group which could include multiple Anatini genera and *Aythya* (Supplementary Note 3). The diversity of ducks alongside the other aquatic birds mirrors that of earlier lakeside communities[65], suggesting that the residents likely had direct access to the lake resources and targeted shallow waters with open grasslands and marshy areas with standing vegetation[66]. At Teotihuacan, this approach provides an improved means to identify lake exploitation practices, evidence for which has been tantalisingly scarce[67,68].

Yeomans et al.[69] presented peptide biomarkers for the identification of Anatidae eggshell and demonstrated the presence of a year-round wetland during the Pleistocene-Holocene transition at the site of Shubayqa in Eastern Jordan. They identified swan (*Cygnus*) and goose (*Anser* or *Branta*) and one unidentified group of ducks. Using dataset 2, we reassessed the four published LC-MS/MS analyses available for these samples and identified multiple consistent SAPs across proteins. These allow us to attribute the previously unidentified group of ducks to either shelduck (*Tadorna* sp.) or Egyptian goose (*Alopochen aegyptiaca*). Alopochen aegyptiaca is less likely because it is not known to occur naturally in Jordan[70]. This taxon cannot be ruled out on a molecular basis because many of its protein sequences are not available in the database, as only a single genome, with low coverage across some eggshell genes, was available. Meanwhile, Tadorna is known to occasionally breed in the region today[70] and the peptides detected in the archaeological sample match the available *Tadorna* protein reference sequences for this taxon (Supplementary Note 3). The *Anser* or *Branta* specimen is now securely identified as *Anser* sp., based on SAPs that consistently separate the two taxa in XCA1,XCA2, OC116 and BPIfoldB4 peptides, although cannot be identified to species given the observed intraspecies variability in *Anser* OC116 sequences. The swan, in contrast, matched uniquely to protein sequences found in mute swan (*Cygnus olor*), where multiple SAPs across XCA2 and OC116 distinguish this species from all other individuals belonging to this genus. Mute swan is considered a rare or accidental visitor to Jordan today[71], making our finding the earliest evidence of this swan breeding in Jordan and further highlighting the potential of palaeoproteomics for palaeoecology.

Our systematic investigation highlighted the widespread presence of SAPs derived from heterozygosity and intraspecific variability both in Anatidae, and non-Anatidae species.

Intraspecific SAPs are most commonly detected in eggshell proteins but were also observed in bone collagen. This means there is an urgent need to expand existing standards for palaeoproteomic analysis[2,72] and improve existing standards for identifying taxon-specific "marker" SAPs for both collagen (as predicted by Richter et al.[1]) and non collagenous proteins.

- Sampling multiple individuals per taxonomic group (i.e., genus or species) is required to distinguish between the effects of several evolutionary mechanisms, including heterozygosity, hybridization, domestication, and artifacts from sequencing, assembly or annotation[33,43,45,73].

- A general trend is developing in the field that markers should be confirmed using both genomic and proteomic data, and that samples from modern reference materials should aim for at least three individuals for a given taxon[18,20,21]. Given that in our admittedly small set of 8 *Anas platyrhynchos* we observed that heterozygous SAPs were frequently close to 50% (Fig. 3), the probability of observing the "most common" mutation at least once would be 87.50% when three individuals are sampled, and 93.75% when four individuals are sampled. For highly variable proteins, the sample population of reference material should also be considered, as characteristic mutations may change geographically and overtime.

- Where available, protein annotation should be carried out using phased genomic data, i.e., when both sets of parental chromosomes are assembled, so that heterozygosity is not hidden[74–77].

- Genomic markers should be considered alongside studies on modern proteins, such as tandem mass spectrometry of proteins extracted from several specimens of reference material, i.e., bone and eggshell[13,78].

- Reference databases and biomarkers for the analysis of ancient samples by mass spectrometry should be accompanied by quality controls, at a minimum indicating the number of individuals used for biomarker confirmation.

- The issues we have discussed here are in many ways similar to the debates surrounding taxonomic determination in zooarchaeology and palaeontology, where many seminal works have proposed measures for rigorous standards in discovering and employing taxonomic markers to identify species or genera of extinct and extant organisms (see especially[79–81]).

Overall, the combination of protein sequence annotation and the assessment of the extent of within- and between-taxon variability carried out in this study support the utility of protein sequences for reconstructing evolutionary relationships when testing very clear hypotheses, rather than directly placing an unknown taxon on a branch of a tree[13,78]. Indeed, the variable topology observed across bone and eggshell protein-based phylogenies highlights that using single or fragmented proteins for phylogenetics needs to be approached with caution. At the same time, the extensive taxonomic coverage of avian protein sequences, covering ~65 % of Anatidae species, provided the means to reliably improve identification of avian taxa from two archaeological sites spanning key periods of human history. This allowed us

to overcome traditional limitations in morphological identification of avian remains due to fragmentation and high species diversity, to examine past ecologies, biodiversity, avian distribution and cultures. However, such studies must rest upon extensive curated protein datasets such as the one presented here, and a higher burden of proof for establishing the biomarkers for taxonomic identification. While here we focused on birds, the issues we highlighted derive from general evolutionary mechanisms and as such are likely to affect palaeoproteomics applied to all life forms, extant and extinct.

## Methods

### Annotating reference protein sequences from avian genomes Genomic dataset

The data included 173 genomes from 112 Anatidae species; 74 were downloaded from NCBI in February 2024[82] and an additional 97 genomes were provided by the Bird 10,000 Genomes Project[31] (B10K, that can be downloaded from https://b10k.com/index/index/species.html). Two additional haploid-resolved assemblies for *Anas platyrhynchos* individual bAnaPla2 were downloaded in May 2024 from the Vertebrate Genome Project[33] (VGP) (now available on NCBI under GCA_964188345.1 and GCA_964188335.1). Genome species and accession list is available in Supplementary Data 1. Species names were modified to be consistent with the Birds of the World digital checklist for 2024[41].

### Reference proteins (queries)

Reference sequences for collagen type I (COL1a1 and COL1a2), c-type lectins (XCA1 and XCA2), Ovocleidin 116 (OC116), albumin, BPI fold containing family B member 4 (BPI-fold-B4, BPIFB4), clusterin, lactadherin, ovalbumin, ovocalyxin 32, ovomucoid, and ovotransferrin were downloaded from NCBI and SwissProt (see Supplementary Note 4, Supplementary Table 5 and Supplementary Fig. 16). Where possible, we used RefSeq annotated proteins for *Anas platyrhynchos* genome (GCF_015476345.1).

### Annotation pipeline

Protein annotation was performed using the University of Turin's High Performance Computing cluster ($C^3S$). The basic annotation of a query protein against a target genome was conducted in two steps using NCBI BLAST+ v.2.16.0[83] and Exonerate v.2.2.0[84]. First, the genome was indexed and a tblastn search with the query protein identified matching scaffolds which were extracted to minimise the search space. We then used the exonerate protein2genome model to align the query protein to the scaffolds to predict and extract "best-hit" coding regions for the query protein, which are then translated with fastatranslate[84].

Additional annotations were performed to improve coverage of protein sequences as sometimes only parts of the sequence could be identified in the genomes. The query was cut into seven overlapping segments so that each part of the protein was covered by two segments. Exonerate was used to align each query segment to the scaffolds to extract and translate matching regions and then filtered to retain only those segments for which a blastp search against the reference protein sequence matched with a similarity greater than 90% for eggshell proteins or 98% for collagen. Filtered segments were aligned

using MAFFT v.7.453[85] to the query sequence and the "best-hit" sequence and consensus sequence was created (excluding the query) using em_cons from EMBOSS v.6.5.7[86]. We found that this consensus sequence often had better coverage than the single "best-hit" sequence and substantially reduced spurious segments that can occur during assembly or automated annotation of genomes[43].

### Curation of annotated proteins

For each protein, an alignment was created using consensus sequences from all genomes, or the "best hit" sequence if no consensus was achieved. The sequences for each genome were then manually curated, and compared against closely related specimens and that specimen's individual fragment alignment, to identify and remove errors, especially frameshifts and incorrect splicing. Where only one annotation was available for a portion of the protein sequence that showed variation, a judgement was made whether to include a SAP or recode the variant as missing (i.e., a gap) based on comparison to other species, the length and type of variation, and its proximity to the end of an exon - a common location for splicing artefacts to occur, which will frequently be observed in multiple other species. Generally, individual SAPs with no clear error were retained, while variants longer than three consecutive SAPs were removed because it was considered unlikely that such extensive variation would occur in proteins from closely related species. All modifications were recorded using Geneious Prime 2024.0 (https://www.geneious.com) and are available in the associated repository[87]. For ovotransferrin, all annotated proteins were trimmed to exclude all amino acids after position 705 in the reference protein (query), retaining the documented section of the protein considered representative of ovotransferrin[88,89], see Supplementary Note 4 for further details.

No protein sequences were recovered from nine genomes, all *Cygnus atratus*. Four genomes were also excluded from the analysis as their protein alignments indicated potential misidentifications of the sample species: D2006016059 ("*Anas acuta*"), DT2207277613-1 ("*Mareca penelope*"), D2104051175 ("*Mareca sibilatrix*"), and D2104051303 ("*Merganetta armata*").

### Assessment of annotation accuracy

In order to assess the validity of our annotations, we compared the results from our annotation pipeline to proteins annotated by NCBI's RefSeq annotation pipeline. Five Anatidae genomes have been annotated by NCBI's RefSeq pipeline, which were also annotated by our pipeline: GCF_015476345.1 (*Anas platyrhynchos*), GCF_009819795.1 (*Aythya fuligula*), GCF_009769625.2 (*Cygnus olor*), GCF_013377495.2 (*Cygnus atratus*), and GCF_011077185.1 (*Oxyura jamaicensis*). Because of the irregular naming often applied to annotated proteins, we ran our reference protein sequences against the RefSeq genomes for Anatidae using blastp[83]. When multiple isoforms were matched, all were included. The RefSeq sequences were aligned to the reference and curated sequences from our study and examined for differences over the curated portion of the sequence, i.e., SAPs, as well as any additional amino acids or sequence coverage present in one sequence but not the other. These were recorded for the most similar sequence when multiple isoforms were available. Some discrepancies identified in the initial assessment of annotations led to the

re-annotation of COL1a1, Clusterin and Ovotransferrin with different or additional reference proteins (see Supplementary Note 4). Final comparison of curated sequences to ReqSeq proteins is presented in Supplementary Note 1, Supplementary Data 6.

### Creation of datasets

After establishing the validity of the annotation pipeline, the curated sequences (Supplementary Data 2–3) were collated to create two subsets of the data for further analysis.

Dataset 1 was created to examine the variability in protein sequences across taxonomic groups, and assess the utility of eachprotein for identification by mass spectrometry. This included 1832 annotated protein sequences that were at least 70% complete, including identical sequences from the same species. Some proteins were also trimmed for analysis, specifically the non-helical regions of COL1a1 and COL1a2 which include additional variability but are not recovered archaeologically, and the first 29 amino acids from the n-terminus of Clusterin, which had poor coverage in the dataset.

Dataset 2 was used as the search database for protein-based identification by mass spectrometry. It contains 1566 sequences from all 13 proteins and retains fragmented or short sequences comprising at least 30% of the full sequence length. For all proteins except ovotransferrin as stated above, sequences retained their original length, excluding likely erroneous portions removed during curation. For each species, this dataset provides an example of each unique sequence identified, with no duplication of identical sequences that may be shared across individuals. Where two sequences from the same species were identical in amino acid sequence but differed in length, the longest sequence was retained. Where two sequences from different species were identical, both sequences were retained.

### Examination of short read sequence data to evaluate intraspecies variants

In order to evaluate the potential biological and non-biological origins of the observed intraspecies variation, we focused on OC116, which showed the highest variation in sequences annotated from 14 *Anas platyrhynchos* genomes. We focused on short-read data produced by Illumina instruments because of their low 0.1–1% sequencing error rate[90]. Nine Sequence Read Archives (SRA) were available for seven of the genomes and were downloaded from NCBI using SRAtoolkit v.3.1.0 (Supplementary Table 1). This includes two SRAs from different biosamples for the *Anas platyrhynchos* RefSeq genome (GCA_015476345.1), which was assembled from pooled individuals. In contrast, GCA_037218355.1 is a genome derived from a single biosample, and two SRAs are available from the same individual. To serve as a target sequence for aligning reads belonging to the *OC116* gene, we downloaded the annotated gene for this protein, NC_051775.1:45251005-45254263 from the NCBI RefSeq genome assembly GCA_015476345.1. One SRA, SRR25181664 corresponding to the genome GCA_030704485.1 was excluded based on the frequent observation of three variants at a single nucleotide location, which is not expected for a diploid organism and may indicate contamination of the genome or assembly from multiple genetically variable individuals.

This resulted in eight SRAs from seven individuals of *A. platyrhynchos* available for analysis.

For analysis of COL1a2 variants, only one SRA was identified that would allow us to examine intraspecies variability for this protein. The SRA for *Anser indicus* (GCA025583725) was examined and aligned to the COL1a2 gene for *Anser cygnoides* (XM_048054476.2).

### Short read coverage of SNPs

The SRA reads were trimmed with Trimmomatic v.0.39[91] (LEADING:3 TRAILING:3 SLIDINGWINDOW:4:15 MINLEN:36) to remove low quality reads and trim low quality sequence ends before alignment to the target gene using the bwa-mem2 v.2.2.1[92] mem function. The aligned reads were then sorted and the reads mapping to the target gene were extracted to a bamfile and indexed using samtools v.3.1.0[93]. Mapped reads were then visually inspected in Integrative Genome Viewer[94] (IGV v.2.17.4) with filters for duplicate and vendor failed reads and flagging mapping quality <30, to examine read coverage and mapping quality of SNP sites that fall within proteincoding regions of the gene for each of the *Anas platyrhynchos* samples. Fair read coverage was considered for 10 mapped reads. The *Anser indicus* sample was examined for read coverage at the nucleotide location corresponding to the one SAP identified in this specimen.

### Gene duplication

To examine for potential duplication of the *OC116* gene that could be causing the observed heterozygosity, the reads mapped in the previous step were extracted and mapped using bwamem2) against the phased genome for *Anas platyrhynchos* bAnaPla2. The assumption is that duplication of the OC116 gene would cause reads to match in multiple places across both haploids. The mapping quality score (MapQ) was used to assess the confidence that a read matches confidently to a given region in the genome, and a MapQ score of three or less was considered evidence that a read matched to more than one area of the genome (Supplementary Table 2).

### Variant calling

Variant calling uses probabilistic models to determine what intraspecific SNPs are well supported for the target gene, and provides information about the frequency of different variants across the sample population[95]. We performed variant calling using GATK v.4.5.0.0[96] following best practice guidelines[97,98]. Using the reads that were previously mapped to the target *OC116* gene, we performed single-sample Genomic Variant Call Format (GVCF) calling for each sample, followed by joint genotyping across all samples. The resulting variants were then filtered to remove those with low confidence values, including mapping quality less than 40 (MQ < 40.0), and filtered again using VCFtools (0.1.16)[99] to exclude variants with read depth less than 10 (DP = < 10) and low minor allele frequency (maf = <0.05). The resulting data provides the location, identity and frequency of SNPs that are most likely to result from true variability among the *Anas platyrhynchos* samples (see Supplementary Note 2).

### Analysis of pairwise distance

A pairwise distance matrix was created for each protein from dataset 1 in R v.4.4.2[100] using the dist.alignment function with the identity method (square-root of the pairwise distance) and ignoring gaps (seqinr v.4.2-36[101]). These matrices were subsetted to include only genera represented by at least five different species. For each protein, intraspecies and interspecies distances were calculated individually for each genus using the sppDist function from the spider package v.1.5.0[102]. The sppDist function was additionally called on the entire OC116 subset to calculate the intergenus distances for this protein. Calculated distances were combined for each protein and summarised using violin plots from the ggplot2 package v.3.5.1[103]. The average number of inter-taxon SAPs for each protein in Fig. 3 and Supplementary Data 4 was calculated as the mean distance, squared, multiplied by the length of the sequence.

### Phylogenetic trees

We reconstructed a maximum likelihood phylogenetic tree for each of the 13 protein alignment from dataset 1 using IQTREE v.2.36[104] with 1000 ultrafast bootstrap replicates[105] after selecting the best-fitting model of sequence evolution[106]. We included sequences from NCBI for *Grus americana* as an outgroup and trimmed these sequences to match the length of the remaining species if necessary. We also analyzed all proteins together in a concatenated analysis while allowing separate partitions for each of the 13 proteins.

Phylogenetic trees for single proteins were plotted in R using treeio v.1.30.0[107], tidytree v.0.4.6[108] and ggtree v.3.14.0[109] packages. Trees were trimmed to exclude the outgroup and scaled individually to improve visibility of branches. The concatenated protein tree was visualised with branch lengths excluded and tip labels aligned. Discrimination of taxa was considered well-supported for bootstrap scores > 80 and highly supported for bootstrap scores > 90.

### Annotation of non-Anatidae specimens

To test for variation outside of Anatidae, bird species with multiple genomes available were downloaded from NCBI and annotated using the pipeline described above for OC116 and Ovocalyxin-32, proteins which both have homologues in mammals (matrix extracellular phosphoglycoprotein (MEPE) and Retinoic acid receptor responder protein 1 (RARRES1) respectively). Our previous testing had identified some species where intraspecies variability had occurred (*Rhynochetos jubatus* - Gruiformes, *Pelecanus crispus* - Pelecaniformes and *Chlamydotis macqueenii* - Otidiformes), while other species were chosen because many genomes were available (*Calidris pugnax* - Charadriiformes, *Grus americana* - Gruiformes, *Lycocorax pyrrhopterus* - Passeriformes, and *Accipiter gentilis* - Accipitriformes) Supplementary Data 7. Thus, this sample represents a very small list, covering only six orders of birds. In order to annotate this diverse group most effectively, we employed multiple OC116 reference sequences and the best were retained, namely *Aphelocoma coerulescens* (XP_068854609.1), *Rissa tridactyla* (XP_054058069.1), *Grus americana* (XP_054681061.1), *Accipiter gentilis* (XP_049670802.1). For ovocalyxin-32, we also employed *Haliaeetus albicilla* (XP_069648969.1) as the reference sequence.

Annotations were curated and aligned as above and the number of locations where intraspecies SAPs occurred were counted and are presented in Supplementary Data 8.

## Application to archeological data

We reanalysed published tandem mass spectrometry (nano LC-MS/MS) data for samples MC148, MC123, MC182 and MC171 from Codlin et al.[19], representative of unidentified duck groups labelled 1-4, respectively (available through ProteomeXchange, PXD034547). These samples were analysed on Peaks Studio 11[110] using dataset 2 combined with all avian proteins available on NCBI as of 01/08/2024 and common contaminants (cRAP: https://www.thegpm.org/crap/). Samples were run using a Peaks Spider search with in-built PTMs and the following settings: Precursor Mass Error Tolerance 15.00 ppm, Fragment Mass Error Tolerance 0.05 Da, Variable Modifications, set for Deamidation (NQ), Hydroxyproline (P), Oxidation (M). Data were filtered using the following: Peptide FDR = 0.5%, Proteins −10 lgP  15.0, Denovo Only ALC  80.0%. Filtered results were searched to confirm peptide sequences for the key distinguishing peptide mass markers identified in Codlin et al.[19], and these peptide sequences were examined in the complete database to observe which species presented these peptides, and which closely related species could have this peptide but are missing this portion of the sequence. These species lists were compared against the top protein hits from Peaks and the species possibly available in pre-Colonial Mexico.

Samples from Yeomans et al.[69] included two from the unknown Anatidae group, PALTO 114D and PALTO 119D, *Cygnus* sp., PALTO 689, and *Anser/Branta* PALTO 693 (available on ProteomeXchange, PXD047233). These tandem mass spectrometry data were searched using PEAKS 11 using the same settings as above, except for removing Hydroxyproline (P) and the addition of Carbamidomethylation (+57.02) as a fixed modification. The filtered proteins were examined to identify which species were identified as top hits, and to confirm quality and coverage of key distinguishing amino acids (Supplementary Data 10–33). See Supplementary Note 3, Supplementary Fig. 16 and Supplementary Tables 3 and 4, for full discussion of identification and MS2s of selected peptides. Identifications were made by examining the range of variation for the SAPs observed in the archaeological samples, giving weight to SAPs that are consistent across a range of individuals or species. In the case of *Cygnus olor*, identification was made despite the presence of only one individual in the database. This is due to the fact there are multiple SAPs shared between the archaeological specimen and the phased genomic data available for *C. olor*. More specifically the identification was made based on matches to SAPs that are either not found in other species of *Cygnus*, or that are shared between *C. olor, C. melancoryphus* and *C. atratus* - the latter two being birds which are native to South America and Australia respectively. In our interpretation about final species identifications, we considered information about current species distribution[66] and the fact that in both Jordan and Central Mexico, the environment has changed considerably since the date of occupation of the archaeological site and thus we considered it possible that species never recorded in these regions could potentially have been found there in the past.

## Reporting summary

Further information on research design is available in the Nature Portfolio Reporting Summary linked to this article.

## Supplementary Material

Refer to Web version on PubMed Central for supplementary material.

## Acknowledgements

We would like to thank the Bird 10,000 Genome Project (B10K, https://b10k.com/) for early access to unpublished Anatidae genomes and the Competence Centre for Scientific Computing at the University of Turin for use of computing resources. This project has received funding from Independent Research Fund Denmark, Research Project 2, Grant 1024-00032B to L.Y., the European Union to B.D. (ERC-2023-COG HORIZON AviArch, 101125532) and to M.C.C. (MSCA-2022-PF Horizon Aviculture, 101110437). The views and opinions expressed, however, are those of the authors only and do not necessarily reflect those of the European Union or the European Research Council. J.S. is supported by a research grant (42153) from VILLUM FONDEN.

## Data availability

All data, curated sequences and related files produced are available in the supplementary files or on Zenodo at https://doi.org/10.5281/zenodo. 16932720[87]. See Supplementary Note 5 for a description of files available on Zenodo. Genomes used in this analysis are publicly available for download from NCBI or the Bird 10,000 Genomes Project website https://b10k.com/index/index/species.html. The complete genome accession list is available in Supplementary Data 1, accessions for Short Read Archives are available in Supplementary Table 1 and for reference proteins accessions are provided in Supplementary Table 5]. LC-MS/MS data used in this article was downloaded from through Proteo-meXchange (PXD034547 and PXD047233). Source Data for Figs. 1–5 can be found on Zenodo at https://doi.org/10.5281/zenodo.16932720[87], as well as in the Supplementary Data 4 for Fig. 3 and Supplementary Data 9 for Fig. 5.

## Code availability

All source code relating to figures, tables and analyses, including a demo for the annotation pipeline, are available on Zenodo at https://doi.org/10.5281/zenodo.16932720[87]. A description of these files is available in Supplementary Note 5.

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

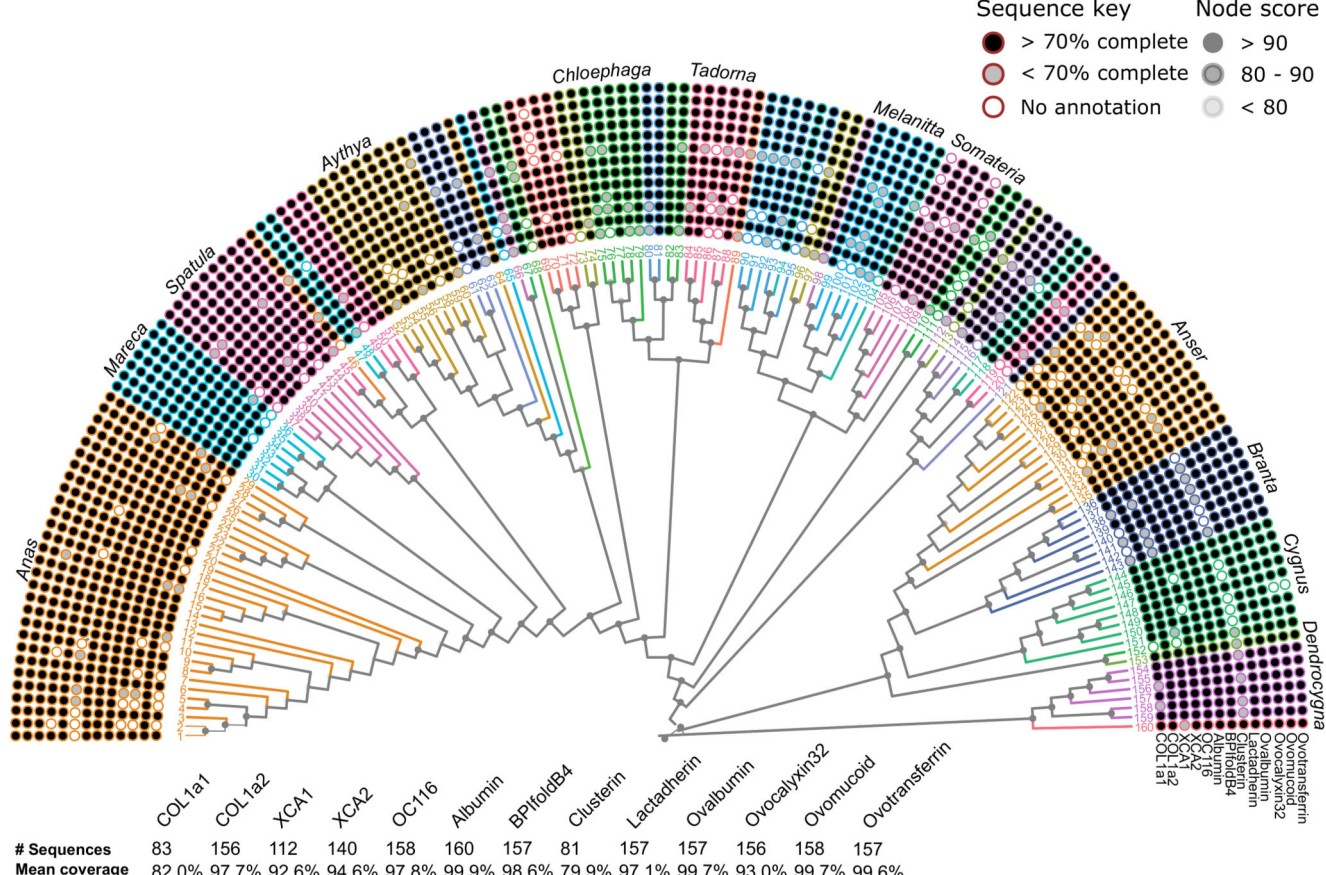

**Fig. 1. Annotation of 13 proteins across 160 Anatidae genomes.**

The phylogenetic tree was constructed from a concatenation of 13 avian proteins showing complete (> 70%), partial (<70%) and missing annotations for the 160 Anatidae genomes and the number of individuals annotated with at least 70% sequence length and mean sequence coverage for each protein. Node score represents the bootstrap resampling support for each node. Samples are coloured by genus and index values represent genomes (see Supplementary Data 5, Supplementary Fig. 1 for species names and genome accession). Genera with at least 5 species are labelled. The tree was rooted with the outgroup *Grus americana* (whooping crane), which is not shown in the plots. Source data available on Zenodo[87].

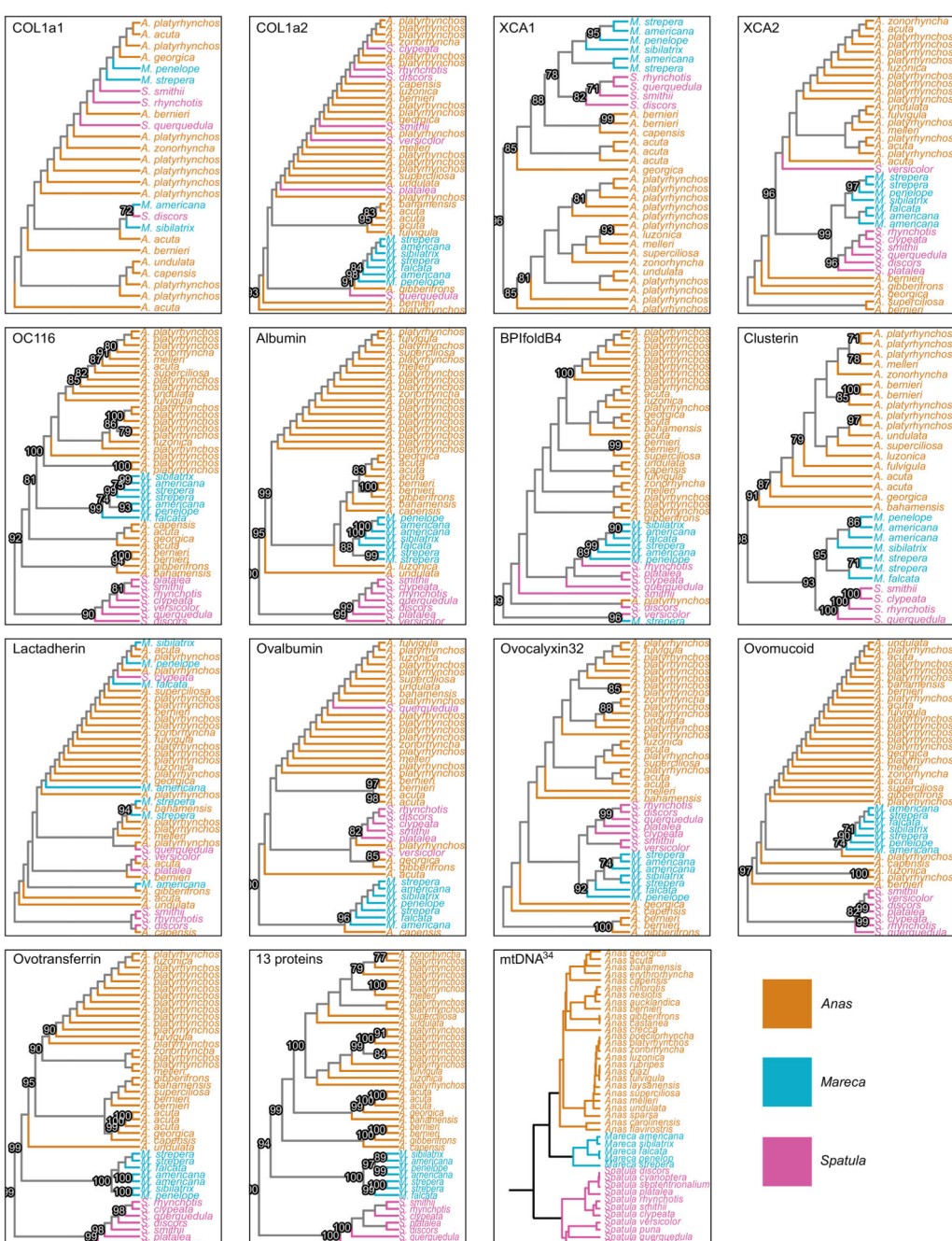

**Fig. 2. Variable topologies of *Anas, Mareca* and *Spatula* seen in individual protein trees.**
Bootstrap scores above 70 are displayed. The trees were rooted with the outgroup *Grus americana* (whooping crane). Also shown is a portion of the mtDNA phylogeny from Sun et al[34]. (CC BY 4.0). The dashed line indicates the branch was modified from the original. Source data available on Zenodo[87].

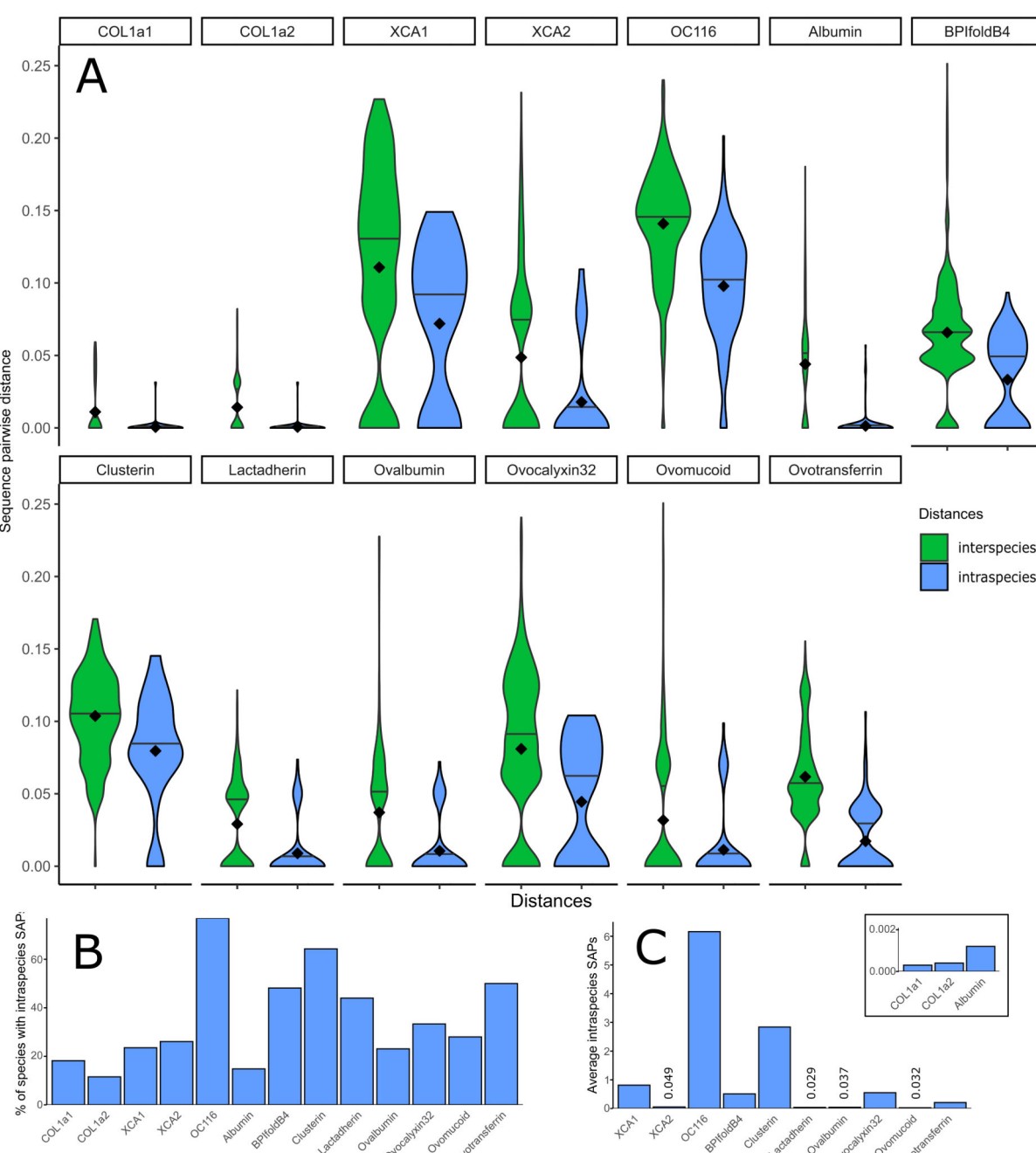

**Fig. 3. Measures of inter and intraspecies variation for 13 avian proteins.**
**A** Summary of pairwise distances between species (in green) and within species (in blue). Black line represents the median distance value, black diamond represents the mean. **B** Percentage of species[mi] where intraspecies SAPs were observed. **C** Mean number of SAPs per pairwise comparison across the entire dataset. Source data available on Zenodo[87] in Supplementary Data 4.

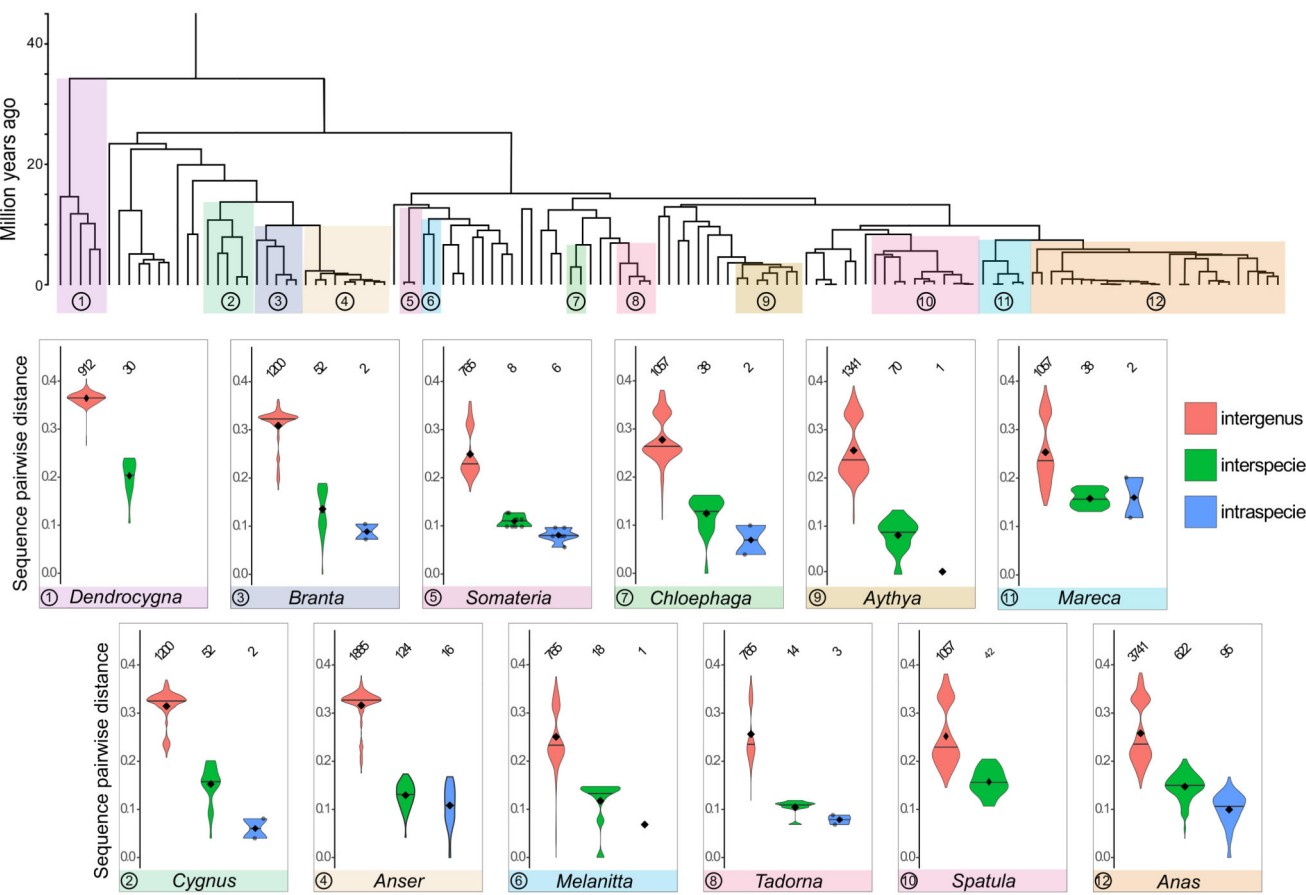

**Fig. 4. Summary of intertaxon pairwise distances for Ovocleidin 116 (OC116), separated by genus.**

Intertaxon distances reflect deep evolutionary divergences, as shown in the phylogenetic tree modified from Fig. 2. Sun et al.[34]. (CC BY 4.0). Density is shown as the width of the plot object. Black line represents the median distance value, black diamonds represent the mean. The numbers represent the number of pairwise comparisons available at each taxonomic level and individual points are shown where sample size is <10. Source data available on Zenodo[87] and in Supplementary Data 9.

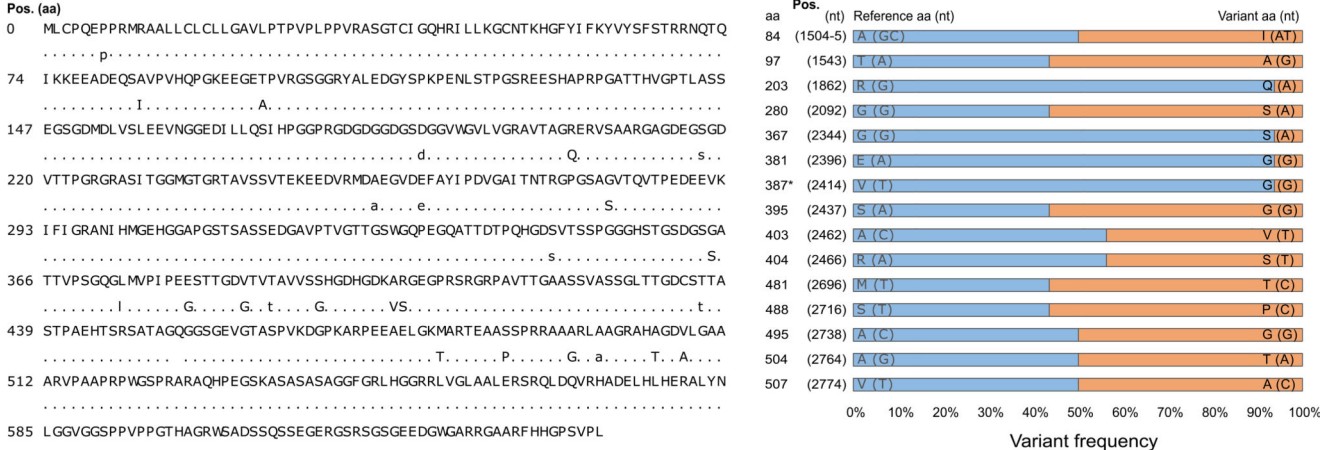

**Fig. 5. Summary of variant calling for OC116 for eight *Anas platyrhynchos* SRAs.**
Left: Alignment of reference OC116 annotated from reference genome of *Anas platyrhynchos* (GCA_015476345.1) and the mutations supported by variant calling. Uppercase letters represent missense mutations while lowercase letters represent silent mutation sites. Right: Occurrence of SNPs causing missense mutations over the 8 SRAs, showing amino acid (aa) positions (pos.) relative to the OC116 reference sequence, and nucleotide (nt) positions relative to the OC116 gene (NC_051772.1:LOC101798662). Amino acids and nucleotides observed in the reference sequence are in blue, while mutations observed in the SRA samples are in orange. Note that the variant at aa pos. 84 is caused by two SNPs. Four additional SAPs were observed in annotated *Anas platyrhynchos* OC116 sequences (Supplementary Fig. 15) that were not evaluated because SRAs were not available. *denotes a variant that was confirmed by variant calling, but is not expressed in the annotated proteins (see Supplementary Note 2). All positions are relative to the reference sequence, not aligned datasets.

