## [Peer Review File · Nature communications]

A library of avian proteins improves palaeoproteomic taxonomic identification and reveals widespread intraspecies variability

Corresponding Author: Dr Maria Codlin

Version 0:

Reviewer comments:

Reviewer #1

(Remarks to the Author)

Codlin et al annotate bone and eggshell proteins sequences from Anatidae from available genomic data. They then use these annotated protein sequences to identify avian remains at two archaeological sites. Importantly, they also show that phylogenetic trees based on protein sequences do not always reflect evolutionary relationships, and that intra-species variation in protein sequences can further complicate such analysis. This study is well-structured, has been executed with a robust methodology, and has significant implications for the field of palaeoproteomics. I recommend publication after addressing some minor comments listed below.

Throughout the manuscript, there are many instances in which binomial nomenclature is used for the taxa that are studied. To reach a more general audience, I believe it would be beneficial to also include the common name when specific taxa are first introduced/mentioned.

Lines 80, 91 & 252: Please define the abbreviations for BPI-fold-B-4, OC116, and aDNA.

Line 111: What does the superscripted mi mean? Perhaps elaborate on this in the text.

Line 161: Here, it is mentioned that there are 26 SNPs in the protein coding region for OC116 in *Anas platyrhynchos*. However, when I look at figure 5, I can only see 24. Is there an error in the text or figure?

Line 297: How many samples were analysed exactly? What defines if a sample is representative?

Lines 294-305: The previously unidentified group of ducks is identified here as *Tadorna* sp. based on the identification of 2 specimens from this group of 15 (Yeomans et al 2024). How confident are the authors that all specimens from this group are of *Tadorna* sp? For the avian remains from Teotihuacan, the reasoning for this is clearly described in the Supplementary Information, but for the unidentified group of ducks from Shubayqa, this is not clearly described. Please elaborate.

Supplementary Table 12: The samples from Teotihuacan are missing from the title of the table.

Some small textual comments:

Line 12: there is a double space between our ancestors.

Line 77: there is a double space between annotated 1928.

Line 250: should it be ancient proteins?

Line 341: overcome should be overcome.

Line 371: remove the space before reference 72.

Line 426: there is a double space between for protein-based.

Line 497: *Grus americana* should be in italics.

Line 502: there is a double space between v.3.14.096 packages.

Line 535: *Cygnus* and *Anser/Branta* should be in italics.

Supplementary Note 3, paragraph 1: there is a double space between respectively. Protein-based.

Supplementary Table 12: *Cygnus* and *Anser/Branta* should be in italics.

(Remarks on code availability)

Reviewer #2

(Remarks to the Author)

The intraspecies protein variation observed in this study is a very exciting result. But I do have concerns about the characterisation of the broader field of palaeoproteomics in this manuscript and what I feel is a mischaracterisation or at least a misunderstanding of the aims and limitations of ZooMS. I understand that the authors want to make clear the relevance and importance of their study but I feel they are discounting over a decade of research to criticise a foundational paper published in 2009. The complexities of making taxonomic identifications using proteins to the highest possible level of discrimination is discussed in several papers. One particularly interesting case study, for instance, is the Jensen et al., 2020 paper (cited below) that identifies peptides which would be useful for taxonomic identification using LC-MS/MS but are not appropriate for identification using ZooMS. The authors encounter similar challenges in their research as they attempt to translate the SAPs identified in their datasets into peptides visible in MALDI-TOF MS spectra.

I understand that the authors are cautioning that SAPs are potentially more frequent within species than had previously been assumed but in doing so they simplify and mischaracterise ZooMS and the broader palaeoproteomic field. I give specific examples below.

I thought the description of the methods and results was fantastic. Everything was described in great detail and will be a welcome addition to the broader palaeoproteomic literature. The description of Zenodo documents in the supplementary was also a nice addition.

Line 45 - Is this a quote? If it is then it should be cited, as I don't think literature supports this statement. As you say later in the introduction (Line 58), SAPs within species have already been encountered in palaeoproteomic literature, so I do not think it is accurate to say that most of the discipline is underpinned by this assumption.

Line 51 - I was unable to access the reference used for this quotation but I think the assertion that bone collagen carries any specific level of taxonomic information is mischaracterised. The lead author of the cited publication has frequently identified species-specific level identifications using ZooMS, including in his foundational 2009 ZooMS paper. ZooMS neither relies on the assumption that bone carries genus-level taxonomic identifications nor is it underpinned by this assumption. It does rely on a certain level of similarity in the collagen (or whichever protein is of interest) but I do not think the authors sufficiently demonstrate that collagen so widely differs between individuals within a species to say that the entire underpinning of ZooMS needs to be called into question.

In fact, species-level identifications in fish have been possible since the on-set of the method, amphibians and reptiles have been shown to have species-level identifications, as have exceptional cases of mammals like the arctic fox and rodents/small mammals.

It is certainly true that the method is frequently described as most commonly achieving genus and family level identifications but this language is used to convey the expectations that should be adopted when attempting to apply ZooMS (rather than shotgun proteomics or aDNA analysis) to samples, especially in the cases of large and medium sized mammals remains. Considerable attention has been paid to identifying new peptide markers that can discriminate between fauna in the same genus to achieve species-specific (or tribe-specific) identifications. Overcoming the initial limitations of the 2009 database has been a central theme of ZooMS research since the development of the P/Cet markers in Buckley et al., 2014.

Examples of species-specific identifications:

Buckley, M., Gu, M., Shameer, S., Patel, S., Chamberlain, A.T., 2016. High-throughput collagen fingerprinting of intact microfaunal remains; a low-cost method for distinguishing between murine rodent bones. *Rapid Commun. Mass Spectrom.* 30, 805–812.

Dierickx, K., Presslee, S., Hagan, R., Oueslati, T., Harland, J., Hendy, J., Orton, D., Alexander, M., Harvey, V.L., 2022. Peptide mass fingerprinting of preserved collagen in archaeological fish bones for the identification of flatfish in European waters. *R Soc Open Sci* 9, 220149.

Harvey, V.L., Daugnora, L., Buckley, M., 2018. Species identification of ancient Lithuanian fish remains using collagen fingerprinting. *J. Archaeol. Sci.* 98, 102–111.

Harvey, V.L., LeFebvre, M.J., deFrance, S.D., Toftgaard, C., Drosou, K., Kitchener, A.C., Buckley, M., 2019. Preserved collagen reveals species identity in archaeological marine turtle bones from Caribbean and Florida sites. *R Soc Open Sci* 6, 191137.

Janzen, A., Richter, K.K., Mwebi, O., Brown, S., Onduso, V., Gatwiri, F., Ndiema, E., Katongo, M., Goldstein, S.T., Douka, K., Boivin, N., 2021. Distinguishing African bovids using Zooarchaeology by Mass Spectrometry (ZooMS): New peptide markers and insights into Iron Age economies in Zambia. *PLoS One* 16, e0251061.

Jensen, T.Z.T., Sjöström, A., Fischer, A., Rosengren, E., Lanigan, L.T., Bennike, O., Richter, K.K., Gron, K.J., Mackie, M., Mortensen, M.F., Sørensen, L., Chivall, D., Iversen, K.H., Taurozzi, A.J., Olsen, J., Schroeder, H., Milner, N., Sørensen, M., Collins, M.J., 2020. An integrated analysis of Maglemose bone points reframes the Early Mesolithic of Southern Scandinavia. *Sci. Rep.* 10, 17244.

Richter, K.K., Wilson, J., Jones, A.K.G., Buckley, M., van Doorn, N., Collins, M.J., 2011. Fish 'n chips: ZooMS peptide mass fingerprinting in a 96 well plate format to identify fish bone fragments. *J. Archaeol. Sci.* 38, 1502–1510.

The statement that the level of taxonomic identification has not been systematically examined is also not necessarily supported and discounts a decade of re-analysis of species included in the ZooMS reference library. The benchmark standard of creating ZooMS reference libraries is to include multiple individuals to avoid potential SAPs (see for instance Janzen et al., 2021 where three individuals per species were used to create reference databases). In instances where variation in the protein of interest is detected within individual specimens of the same species, these peptides are not used

in ZooMS identifications. Since the mass spectrometers used for ZooMS analysis are not capable of the resolution achieved with LC-MS/MS, the peptides used for taxonomic identification need to be observed reliably. SAPs are therefore excluded so that the results can be considered reliable without the use of LC-MS/MS.

As you also identify in your own research in the case of COL1, the proteins generally used for ZooMS identification show higher levels of conservation overall as SAPs are observed far less frequently than in other proteins. If your research had demonstrated that SAPs were rampant in COL1 then this would be an important call-to-arms, but 5 individual SAPs were identified across dozens of specimens. A single SAP within COL1 is also unlikely to completely change the taxonomic identification of a specimen as ZooMS identifications are based on a series of markers and peptides.

Line 52 - As ZooMS has grown, the same species have been re-studied multiple times. For instance the original database (Buckley et al., 2009) was republished and updated in Welker et al., 2016 alongside collagen sequences. Since then, many of those species have had multiple individuals re-analysed to try and identify new peptides for greater levels of identifications.

The COL1 sequences of sheep for instance have been studied as part of Buckley et al., 2009, Welker et al., 2016, Janzen et al., 2021 and in:

Coutu, A.N., Taurozzi, A.J., Mackie, M., Jensen, T.Z.T., Collins, M.J., Sealy, J., 2021. Palaeoproteomics confirm earliest domesticated sheep in southern Africa ca. 2000 BP. *Sci. Rep.* 11, 6631.

Line 55 - ZooMS does not rely on groupings like genera, it uses this language to simplify discussion of the method (by saying things like genus or family level identifications) but the taxonomic groupings it can achieve frequently do not follow genus groupings. In instances in which fauna can be differentiated from other members of the same genera, this is discussed in the literature. Brown et al., 2021 for instance re-classifies their groupings as "ZooMS taxons" as their identifications do not follow zooarchaeological categories previously identified at the site of interest.

The oversimplification in my view, is in the manner in which ZooMS is discussed in your introductory paragraphs and not in the way that other researchers in the field are applying the method - particularly those working on improving reference databases. See for example:

Jensen, T.Z.T., Sjöström, A., Fischer, A., Rosengren, E., Lanigan, L.T., Bennike, O., Richter, K.K., Gron, K.J., Mackie, M., Mortensen, M.F., Sørensen, L., Chivall, D., Iversen, K.H., Taurozzi, A.J., Olsen, J., Schroeder, H., Milner, N., Sørensen, M., Collins, M.J., 2020. An integrated analysis of Maglemose bone points reframes the Early Mesolithic of Southern Scandinavia. *Sci. Rep.* 10, 17244. (Which includes a lengthy supplementary on the complexity of identifying peptide markers suitable for ZooMS analysis)

Janzen, A., Richter, K.K., Mwebi, O., Brown, S., Onduso, V., Gatwiri, F., Ndiema, E., Katongo, M., Goldstein, S.T., Douka, K., Boivin, N., 2021. Distinguishing African bovids using Zooarchaeology by Mass Spectrometry (ZooMS): New peptide markers and insights into Iron Age economies in Zambia. *PLoS One* 16, e0251061. (Which discusses several markers commonly used in ZooMS analysis that should be considered unreliable and highlights the complexities of identifying peptide markers suitable for ZooMS analysis)

Dierickx, K., Presslee, S., Hagan, R., Oueslati, T., Harland, J., Hendy, J., Orton, D., Alexander, M., Harvey, V.L., 2022. Peptide mass fingerprinting of preserved collagen in archaeological fish bones for the identification of flatfish in European waters. *R Soc Open Sci* 9, 220149.

Korzow Richter, K., McGrath, K., Masson-MacLean, E., Hickinbotham, S., Tedder, A., Britton, K., Bottomley, Z., Dobney, K., Hulme-Beaman, A., Zona, M., Fischer, R., Collins, M.J., Speller, C.F., 2020. What's the catch? Archaeological application of rapid collagen-based species identification for Pacific Salmon. *J. Archaeol. Sci.* 116, 105116.

Peters, C., Richter, K.K., Manne, T., Dortch, J., Paterson, A., Travouillon, K., Louys, J., Price, G.J., Petraglia, M., Crowther, A., Boivin, N., 2021. Species identification of Australian marsupials using collagen fingerprinting. *R Soc Open Sci* 8, 211229.

Line 286 and Supplementary: I think it is important in the main text to say that you were able to refine the identifications made in the Codlin paper through the use of LC-MS/MS and not MALDI-TOF. In the supplementary, you describe not being able to use the SAPs identified in this manuscript to refine the identifications from the Codlin paper through peptides which are visible in MALDI-TOF spectra but this is not as clear in the main text. Most people familiar with the field will accept that LC-MS/MS data will generally return a higher degree of taxonomic resolution than MALDI-TOF MS.

Line 311: I think it is important to note here that such standards have been called for (as the authors indicate by referencing Richter's recent review) in previous literature and are already being followed by many people working in the field. Although I appreciate such a direct list of standards the field should adopt as a whole

(Remarks on code availability)

All of the data is made available. A really wonderful example of data sharing!

Reviewer #3

(Remarks to the Author)

This is a great paper that highlights the need for more in depth assessment of proteomic data acquired from archaeological material, when making taxonomic identifications. The paper is well written, easy to follow and all the data has been critically examined with well-reasoned and clear results. The authors make several recommendations which are well justified and should be embraced (where possible) by the field of paleo proteomics.

All the data has been clearly reported and is accessible

I only have two minor comments:

Line 283 – consider having a subheading for the arch case study to make this clearer in the discussion

Line 341 – typo for overcome

(Remarks on code availability)

The access to the dataset was restricted until publication. Only the metadata was available to view

Reviewer #4

(Remarks to the Author)

Mz primary expertise is in the field of proteomics and analytical sciences, and as such my review does not address the construction of phylogenetic trees in great detail.

From my point of view, the manuscript is scientifically sound, with a well-detailed methodology and rigorous data-analysis, grammatically well-written, and addresses some very relevant questions in the field of paleoproteomics. Even though it has been argued that proteomics is a tool for identifying the protein and not necessarily the species, paleoproteomic studies are increasingly used to address species-specific archaeological questions, and as such this manuscript is relevant in trying to analyse some of the challenges for using proteomics for such purpose.

However, I do have a few questions regarding the manuscript.

Very broadly, the manuscript (including the abstract), talks about the widespread evidence for SAPs occurring within species. However, from the data it appears that the vast majority of the SAPs observed are concentrated on eggshell proteins. As such, can the authors please specify this in the abstract and the introduction while discussing the presence of SAPs in general?

To me, the two main findings of the manuscript can occasionally appear somewhat paradoxical and contradictory. Whereas the discussion about intraspecies diversity warns about the challenges regarding the use of SAPs for taxonomic distribution, the archaeological case studies mentioned introduces new reference sequence and uses them for taxonomic identification to genus and species level based on the presence of SAPs. Although some of this is discussed in the Supplementary text (page 7 and 8), it is not sufficiently clear to me how the authors distinguish between the SAPs they use for taxonomic identification of the various Anatidae species with SAPs possibly arising as a result of the reported intra/interspecies variation. Can the authors please clarify this, and perhaps introduce a separate paragraph in the discussion section addressing this point in sufficient detail?

Can the authors please briefly discuss that the mass change corresponding to how many (if any) of the SAPs they observe can coincide with mass changes due to various observed PTMs in the mass spectra? (For e.g. the mass change due to an Q → R substitution can coincide with the mass change due to formylation, or the mass change due to P → Q substitution can coincide with a double oxidation or the oxidation of methionine to sulphone, etc.). Although not directly related to the manuscript, I believe that this will be helpful when it comes to interpreting tandem mass spectrometric data for species identification purposes.

Line 82. Was the value of 70% chosen arbitrarily, or based on previous work/ existing convention?

Line 124. I think this line is very important, particularly given that single amino acid polymorphisms of collagen have been previously used to argue for taxonomic identification of ancient human remains (Chen et al. 2019). Although possibly outside the scope of this paper, it will be very interesting to see if this trend of SAPs hold on mammalian/primate species, given that the majority of the SAPs observed here seem to be associated with proteins involved in egg-laying and, more generally, reproduction.

Line 177. From supplementary table 4, it appears that OC116 has the highest number of SAPs based on mean pairwise by far among (6,4 vs 2,9) all the proteins considered for Anatidae. Can the authors please perform the annotation for the seven additional species described here using a protein other than OC116, for e.g. ovotransferrin or albumin)?

Line 184. Given that collagen is the protein most often used for taxonomic identification and phylogenetic studies based on proteomics, this section should be emphasized and discussed in greater detail. To me, this section reads that the presence of SAPs in collagen is poorly supported, and if that is correct, the effect of SAPs on collagen-based taxonomic identification will be minimal, and the manuscript needs to reflect that. In my opinion, the discussion section needs to have an additional paragraph discussing that importance of the protein (and indirectly, substrate) used for proteomics-based taxonomic identification- from my understanding, the proteins involved in eggshells and/or other reproductive traits seem to be substantially more affected by SAPs as compared to collagen. If that understanding is correct, some of the primary findings of the paper, although extremely relevant to eggshell-based proteomics, are perhaps less relevant for proteins extracted from other substrates like bones, and this needs to be explicitly mentioned in the manuscript.

Line 308. Please can the authors clarify this statement, given that the presence of SAPs in collagen are much lower as compared to eggshell proteins (acc to Supp. Table 4) and previously in line 184 they state at least at least some of the SAPs observed in collagen is poorly supported?

(Remarks on code availability)

I have reviewed the protein analysis of the code in detail, but the construction of the phylogenetic trees are outside my immediate area of expertise.

Version 1:

Reviewer comments:

Reviewer #1

(Remarks to the Author)

I would like to thank the authors for their thoughtful responses to the reviewer questions. I do not have any further comments. This is a wonderful study, ready for publication.

(Remarks on code availability)

Reviewer #2

(Remarks to the Author)

After reviewing the response document and the text files, I have no further edits or changes to request. I appreciate the authors extensive replies to my initial concerns, even in stances when we disagree. I feel the changes to the text are more than appropriate and I look forward to reading the paper once it is published.

(Remarks on code availability)

Reviewer #4

(Remarks to the Author)

I am happy with the authors' responses and clarifications to my questions and concerns, and as such am happy to recommend the manuscript for publication.

(Remarks on code availability)

Response to reviewers:

We thank the reviewers for their thoughtful comments which have improved our manuscript. The main concerns raised by the reviewers are broadly divided into 4 categories:

- 1) Minor clarifications on the text, which we have addressed in full (details below)
- 2) Our characterisation of ZooMS and aspects of the broader palaeoproteomic field - for which we welcome the opportunity to clarify our intent and revise the manuscript to better reflect the complexities and contributions of previous research in this area.

We have rephrased many sections of the manuscript to more carefully describe ZooMS applications and make reference to taxa where species-level identifications have been possible. We have also amended the text to give additional credit to researchers who have contributed to the development of the technique and current protocols for analysis, as well as previous works that have called for more rigorous marker development. (Note that we have used citations here with author-date to better highlight the new references included, while in text these have been included as numbers following the manuscript citation style.)

- 3) The somewhat contradictory outcomes of observing widespread SAPs, which can both confound and improve identification.

We have revised the main text and supplementary materials to more clearly outline how we identify the archaeological taxa using our new protein database.

- 4) The relevance of these findings beyond avian eggshells, especially to collagen (the standard "ZooMS" technique).

We argue that the current protocols for ZooMS and palaeoproteomics in general are not standardised and demonstrate how our findings are applicable broadly to many proteins, including collagen, and to non-avian taxa. We have revised the text to emphasise that, especially whenever non-avian collagen appears to have greater *interspecies* variability, researchers should examine the occurrence of *intraspecies* variability. Additionally, we highlight that many eggshell proteins have homologues in humans, and that variability has also been observed in those homologues, demonstrating that these findings are not restricted to eggshell proteins or avian species. As suggested by reviewer 4, we also annotated non Anatidae genomes for Ovocalyxin 32 to provide additional evidence of intraspecies variation outside Anatidae.

In addition, we modified Figure 3c and Supplementary Table 8 as rounding of data and scale issues prevented variability in multiple proteins from being clearly evident in the graph. We have now added labels to the graph for four proteins with mean SAPs below 0.1, and provide an inset map to demonstrate the SAPs for proteins with means below 0.002. Supplementary table 4 has been updated to reflect non-rounded values. We have also updated Figure 4 to provide individual points on the violin plots where $n < 10$ to meet data presentation standards.

Overall, the aim of our paper is to highlight that intraspecies variability is more common than we thought, and it occurs in the proteins that the palaeoproteomics community has been using to make taxonomic identifications, on which evolutionary and archaeological inferences are made. Our intent is to provide clear guidelines for how to navigate this issue, because while it may confound identifications when reference datasets are limited, it also presents an opportunity for new lines of evidence for studying population dynamics and evolutionary histories.

Reviewer #1 (Remarks to the Author):

Codlin et al annotate bone and eggshell proteins sequences from Anatidae from available genomic data. They then use these annotated protein sequences to identify avian remains at two archaeological sites. Importantly, they also show that phylogenetic trees based on protein sequences do not always reflect evolutionary relationships, and that intra-species variation in protein sequences can further complicate such analysis. This study is well-structured, has been executed with a robust methodology, and has significant implications for the field of palaeoproteomics. I recommend publication after addressing some minor comments listed below.

MINOR COMMENTS

- 1) Throughout the manuscript, there are many instances in which binomial nomenclature is used for the taxa that are studied. To reach a more general audience, I believe it would be beneficial to also include the common name when specific taxa are first introduced/mentioned.**

RESPONSE: We have updated the manuscript to consistently provide common names to the first mention of taxonomic groups except for most genera, for which the group cannot be easily described using a single common name.

- 2) Lines 80, 91 & 252: Please define the abbreviations for BPI-fold-B-4, OC116, and aDNA.**

RESPONSE: We thank the reviewer for pointing this out. We have updated the description of proteins (line 85) to include "Ovocleidin116 (OC116), Albumin, BPI fold containing family B, member 4 (BPI-fold-B-4)" and removed the "a" from aDNA on line 252 since the word "ancient" is already present in the sentence.

- 3) Line 111: What does the superscripted mi mean? Perhaps elaborate on this in the text.**

RESPONSE: The reviewer refers to the statement "13 of 27 species^{mi} where multiple individuals were available for comparison". We have updated the text (line 117) to make this definition clearer. "(13 of 27 species for which multiple individuals were available for comparison, henceforth denoted by "species^{mi}")"

- 4) Line 161: Here, it is mentioned that there are 26 SNPs in the protein coding region for OC116 in *Anas platyrhynchos*. However, when I look at figure 5, I can only see 24. Is there an error in the text or figure?**

RESPONSE: We thank the reviewer for this observation. There was indeed a silent mutation missing from the 6th amino acid position in Figure 5 which has now been corrected. Additionally, two SNPs are responsible for one SAP. We have updated the text to include "..16 of which are responsible for 15 missense mutations..." (line 173) and figure 5 caption "Note that the variant at aa pos. 84 is caused by two SNPs" to clarify this.

5) Line 297: How many samples were analysed exactly? What defines if a sample is representative?

RESPONSE: We have updated the text both here and for the section on data from Teotihuacan to make it clear we analysed published LC-MS/MS data that was available for Anatidae taxa from these previous projects. In both cases this was four samples. These sections now read:

“We reevaluated the four specimens representative of these groups, for which both LC-MS/MS and MALDI-TOF MS data was available, and used dataset 2 to refine identifications...” and “Using dataset 2, we reassessed the four published LC-MS/MS datasets available for these samples...” (line 317)

6) Lines 294-305: The previously unidentified group of ducks is identified here as *Tadorna* sp. based on the identification of 2 specimens from this group of 15 (Yeomans et al 2024). How confident are the authors that all specimens from this group are of *Tadorna* sp? For the avian remains from Teotihuacan, the reasoning for this is clearly described in the Supplementary Information, but for the unidentified group of ducks from Shubayqa, this is not clearly described. Please elaborate.

RESPONSE: While we cannot be 100% certain that all samples from this group of specimens are *Tadorna* sp., we are confident that this is the most likely scenario given that currently *Tadorna* is native to Jordan and occasionally breeds there, while *Alopochen* is considered invasive, and there is no other zooarchaeological evidence to suggest that *Alopochen* was present in Jordan, or nearby regions during the late Pleistocene. We rephrased the supplementary material, adding two new citations and an additional reference to the bird bone assemblage where *Tadorna* bones were identified to clarify and support this position:

“Today, *Alopochen* is considered an introduced species in Jordan, while two species of *Tadorna* are naturally occurring (Andrews, 1995; Meinertzhagen, 1935). While we cannot rule out a change in the breeding and distribution ranges of *Alopochen* and *Tadorna ferruginea* from the Late Pleistocene to today, given that *Tadorna tadorna* occasionally breeds in the region (Andrews, 1995; Wallace, 1983) and specimens from this genus were identified in the osteological assemblage from Shubayqa (Yeomans et al., 2024), *Tadorna* sp. is the best candidate for the late Pleistocene breeding population.” (SI, line 203)

7) Supplementary Table 12: The samples from Teotihuacan are missing from the title of the table.

RESPONSE: We thank the reviewer for pointing this out and have updated the table caption accordingly.

8) Some small textual comments:

Line 12: there is a double space between our ancestors.

Line 77: there is a double space between annotated 1928.

Line 250: should it be ancient proteins?

Line 341: overcome should be overcome.

Line 371: remove the space before reference 72.

Line 426: there is a double space between for protein-based.

Line 497: *Grus americana* should be in italics.

Line 502: there is a double space between v.3.14.096 packages.

Line 535: *Cygnus* and *Anser/Branta* should be in italics.

Supplementary Note 3, paragraph 1: there is a double space between respectively. Protein-based.

Supplementary Table 12: *Cygnus* and *Anser/Branta* should be in italics.

RESPONSE: We thank the reviewer for their careful reading of our manuscript and have made all changes suggested.

Reviewer #2 (Remarks to the Author):

- 1) The intraspecies protein variation observed in this study is a very exciting result. But I do have concerns about the characterisation of the broader field of palaeoproteomics in this manuscript and what I feel is a mischaracterisation or at least a misunderstanding of the aims and limitations of ZooMS. I understand that the authors want to make clear the relevance and importance of their study but I feel they are discounting over a decade of research to criticise a foundational paper published in 2009. The complexities of making taxonomic identifications using proteins to the highest possible level of discrimination is discussed in several papers. One particularly interesting case study, for instance, is the Jensen et al., 2020 paper (cited below) that identifies peptides which would be useful for taxonomic identification using LC-MS/MS but are not appropriate for identification using ZooMS. The authors encounter similar challenges in their research as they attempt to translate the SAPs identified in their datasets into peptides visible in MALDI-TOF MS spectra. I understand that the authors are cautioning that SAPs are potentially more frequent within species than had previously been assumed but in doing so they simplify and mischaracterise ZooMS and the broader palaeoproteomic field. I give specific examples below. I thought the description of the methods and results was fantastic. Everything was described in great detail and will be a welcome addition to the broader palaeoproteomic literature. The description of Zenodo documents in the supplementary was also a nice addition.

RESPONSE: we welcome the opportunity to clarify our intent and revise the manuscript to better reflect the complexities and contributions of previous research in this area, as described in detail below

- 2) **Line 45 - Is this a quote? If it is then it should be cited, as I don't think literature supports this statement. As you say later in the introduction (Line 58), SAPs within species have already been encountered in palaeoproteomic literature, so I do not think it is accurate to say that most of the discipline is underpinned by this assumption.**

RESPONSE: We clarify that this is our argument, and not a quote. Overall, our aim here is to understand the fundamental biological issue of protein variation per se, rather than to

explore the different levels of resolution offered by LC-MS/MS versus ZooMS by MALDI-TOF-MS, which is an analytical/technical issue.

We respectfully disagree with the reviewer's position here and stand by our comment, previously on line 45 which states, "Most applications of palaeoproteomics (the study of ancient proteins by mass spectrometry) are underpinned by the assumption that *protein sequences do not vary within species but only at the level of species or above.*" This assumption, we think, can be perceived at two levels:

a) Within the palaeoproteomics community: as we will argue in later responses, many researchers are still not sampling multiple individuals per species, and even when they are, publications frequently make no distinction between biomarkers established with multiple individuals compared to a single individual when presenting the biomarker results, instead, this information is often hidden in supplementary material. We concede that we have oversimplified our brief mention of ZooMS in this introductory paragraph and that this is unfair to the countless researchers who have worked to improve the application of the method. We have updated the text (see below) to reflect this.

b) Within the community of potential users of palaeoproteomic data (i.e., not the researchers establishing markers, but those who wish to use them), a common question—often raised at conferences—is the taxonomic level at which ZooMS or protein sequences can be used (the two are often conflated). In other words, users seek a rule of thumb to determine whether protein sequences vary at the species, genus, or family level. To date, the lack of multiple genomes per species has led to the inability of assessing this fundamental fact for many taxa. Paleoproteomics researchers have often erred on the side of caution but the technique has been adopted enthusiastically, at a faster pace than the availability of modern genomes.

3) Line 51 - I was unable to access the reference used for this quotation but I think the assertion that bone collagen carries any specific level of taxonomic information is mischaracterised.

RESPONSE: This comment is in reference to our sentence "Additionally, peptide fingerprinting applications such as "ZooMS - zooarchaeology by mass spectrometry" rely on the linked assumption that bone collagen usually carries genus-level taxonomic information¹⁹". Our citation relates to the sentence "Focusing on both domestic and wild vertebrate prey, the review of collagen fingerprinting provided here has demonstrated the ability of this method to distinguish taxa in most cases at the genus level where there is at least ca. five million year divergence, and even at the species level for some large mammal genera (e.g., Camelus)." from Buckley (2018, p. 240).

We have rephrased the sentence as follows:

Peptide fingerprinting applications such as "ZooMS - zooarchaeology by mass spectrometry" have found that, with some exceptions, bone collagen from mammals typically allows for taxonomic discrimination at the *genus level*" (Buckley, 2018). (line 47)

We have also rephrased our introduction to clarify our argument and give more credit to previous research on refining taxonomic resolutions:

While extensive research has continued to refine both the resolution and the confidence of

collagen-based taxonomic identifications across multiple taxa (Buckley & Cheylan, 2020; Codlin et al., 2022; Dierickx et al., 2022; Harvey, LeFebvre, et al., 2019; Janzen et al., 2021; Korzow Richter et al., 2020; Peters et al., 2021; Welker et al., 2016), including species-level identifications of fish (Dierickx et al., 2022; Korzow Richter et al., 2020) and amphibians (Buckley & Cheylan, 2020), the possibility of intraspecies diversity in protein sequences has not been systematically examined by the palaeoproteomics community. (line 50)

- 4) **The lead author of the cited publication has frequently identified species-specific level identifications using ZooMS, including in his foundational 2009 ZooMS paper. ZooMS neither relies on the assumption that bone carries genus-level taxonomic identifications nor is it underpinned by this assumption. It does rely on a certain level of similarity in the collagen (or whichever protein is of interest) but I do not think the authors sufficiently demonstrate that collagen so widely differs between individuals within a species to say that the entire underpinning of ZooMS needs to be called into question. In fact, species-level identifications in fish have been possible since the on-set of the method, amphibians and reptiles have been shown to have species-level identifications, as have exceptional cases of mammals like the arctic fox and rodents/small mammals.**

RESPONSE: We appreciate the reviewer's point of view that this generalisation masks the complexity of this technique and work of other scholars who have investigated taxonomic resolution of ZooMS and have addressed these comments in our previous two responses. We would like to reiterate that our argument is that while *interspecies* differences have been observed and reported, *intraspecies* variation is generally assumed to be absent or at least not widespread, and it is this assumption that has not been systematically tested - due to the lack of multiple individual genomes per species We are not questioning the validity of the ZooMS technique, but highlighting the need for more careful consideration of markers by both those identify them and those who employ them.

Our focus is not on ZooMS per se but on the inherent variability of the protein sequences from which SAPs markers are derived. This is the parameter we investigate when exploring the issue of taxonomic resolution, and for this we look at multiple individuals per species, and multiple species per higher taxonomic grouping. This may well be a mute point in some applications - and may lead to some confusion on the use of the term "species-specific". For example, from the papers mentioned by Reviewer 2, the markers for turtles (Harvey et al. 2019) or those from Jensen et al. (2020) represent species from monospecific genera, or instances where other species are ruled out based on size or geography. In these cases, the markers are defined "species-specific" in practice, but without an assessment of the extent of biological variability.

We have clarified our position by stating clearly that "...the possibility of intraspecies diversity in protein sequences has not been systematically examined by the palaeoproteomics community." (line 52)

To stress the importance of our findings to ZooMS in particular, we have also revised our results to highlight to the fact that intraspecies variation in collagen is not rare: “Mean SAPs may seem negligible for some proteins, including collagen and albumin(Figure 3c), but Figure 3b shows that even for these proteins, intraspecies SAPs were observed for 11-18 % of species^m, hence intraspecies variation is certainly not a rare occurrence and cannot be dismissed.” (line 123).

With this finding, we do not seek to undermine ZooMS, but highlight the very real possibility that researchers may encounter intraspecies variability and that this may impact how researchers plan biomarker development or palaeoproteomic applications addressing hybridization or evolutionary histories. Indeed this may be even more important for ZooMS applications to taxa that do appear to have more *interspecies* variation, like fish and amphibians. For these taxa, our findings suggest the possibility that intraspecies variation may be more common than it is for Anatidae. We think that until a systematic study of intraspecies variation in mammals and other taxa is conducted, we cannot assume that it is rare and that it does not affect biomarker discovery or identification.

We have updated our text to reflect this (line 254):

“Given that some, if rare, intraspecies variation does occur within collagen sequences of birds, as has been observed in orangutans and hominins (Kubat et al. 2023; Tsutaya et al. 2025), the application of collagen peptide mass fingerprinting (ZooMS) to identify archaeological samples should also consider this possibility when identifying biomarkers for taxonomic determination, particularly if the aim is to distinguish closely related groups. This is especially important for fish (Dierickx et al. 2022; Korzow Richter et al. 2020) and amphibians (Buckley and Cheylan 2020) for which interspecific variability is common.”

5) It is certainly true that the method is frequently described as most commonly achieving genus and family level identifications but this language is used to convey the expectations that should be adopted when attempting to apply ZooMS (rather than shotgun proteomics or aDNA analysis) to samples, especially in the cases of large and medium sized mammals remains. Considerable attention has been paid to identifying new peptide markers that can discriminate between fauna in the same genus to achieve species-specific (or tribe-specific) identifications. Overcoming the initial limitations of the 2009 database has been a central theme of ZooMS research since the development of the P/Cet markers in Buckley et al., 2014.

Examples of species-specific identifications:

- **Buckley, M., Gu, M., Shameer, S., Patel, S., Chamberlain, A.T., 2016. High-throughput collagen fingerprinting of intact microfaunal remains; a low-cost method for distinguishing between murine rodent bones. Rapid Commun. Mass Spectrom. 30, 805–812.**
- **Dierickx, K., Presslee, S., Hagan, R., Oueslati, T., Harland, J., Hendy, J., Orton, D., Alexander, M., Harvey, V.L., 2022. Peptide mass fingerprinting of preserved collagen in archaeological fish bones for the identification of flatfish in European waters. R Soc Open Sci 9, 220149.**

- Harvey, V.L., Daugnora, L., Buckley, M., 2018. Species identification of ancient Lithuanian fish remains using collagen fingerprinting. *J. Archaeol. Sci.* 98, 102–111.
- Harvey, V.L., LeFebvre, M.J., deFrance, S.D., Toftgaard, C., Drosou, K., Kitchener, A.C., Buckley, M., 2019. Preserved collagen reveals species identity in archaeological marine turtle bones from Caribbean and Florida sites. *R Soc Open Sci* 6, 191137.
- Janzen, A., Richter, K.K., Mwebi, O., Brown, S., Onduso, V., Gatwiri, F., Ndiema, E., Katongo, M., Goldstein, S.T., Douka, K., Boivin, N., 2021. Distinguishing African bovids using Zooarchaeology by Mass Spectrometry (ZooMS): New peptide markers and insights into Iron Age economies in Zambia. *PLoS One* 16, e0251061.
- Jensen, T.Z.T., Sjöström, A., Fischer, A., Rosengren, E., Lanigan, L.T., Bennike, O., Richter, K.K., Gron, K.J., Mackie, M., Mortensen, M.F., Sørensen, L., Chivall, D., Iversen, K.H., Taurozzi, A.J., Olsen, J., Schroeder, H., Milner, N., Sørensen, M., Collins, M.J., 2020. An integrated analysis of Maglemose bone points reframes the Early Mesolithic of Southern Scandinavia. *Sci. Rep.* 10, 17244.
- Richter, K.K., Wilson, J., Jones, A.K.G., Buckley, M., van Doorn, N., Collins, M.J., 2011. Fish 'n chips: ZooMS peptide mass fingerprinting in a 96 well plate format to identify fish bone fragments. *J. Archaeol. Sci.* 38, 1502–1510.

RESPONSE: We appreciate the reviewer's concern about how we initially presented ZooMS in oversimplified terms and do not disagree that researchers have invested a lot of effort in refining the taxonomic resolution of ZooMS for many taxa. We would like to discuss how some of our concerns are evident in the papers mentioned by the reviewer. We do not wish to criticise these authors as it is clear that the field of ZooMS has progressed greatly over the the past decade and a half, but we think that they provide an exemplary case of the variety of approaches that have been employed by researchers so far, and strengthen the case for the need of tackling this in a standardised manner

First of all, we would like to acknowledge that Dierickx et al. (2022) and Janzen et al. (2021) represent good models for species-specific biomarker discovery as defined by our criteria, while Jensen et al. (2020) also provides a good model for refining biomarkers for distinguishing two specific co-occurring taxa. These studies employ multiple individuals per species, alongside confirmation using LC-MS/MS as well as using available genomic data to annotate and compare theoretical collagen sequences to the proteins physically recovered from bone.

However, Buckley et al. (2016), Harvey et al. (2018) and Harvey et al. (2019) do not specify how many individuals or use only a single individual as a reference specimen for establishing biomarkers (with the exception for one species from Harvey et al. 2019). This includes biomarkers which are supposedly species-specific. These highlight the assumption that a single sample is representative of the species and need to be confirmed with further testing. While many studies are now sampling multiple individuals per taxon of interest, many of the original biomarkers from Buckley et al. (2009) and Buckley and Kansa (2011) are used without question, and the majority of the markers published in the first 5 years were based on single individuals only. As the reviewer has stated, many of these have in fact been replicated and confirmed. But not all of them.

The distinction between *Rattus rattus* and *R. norvegicus* can be used as an example, as it is based on a single marker published by Buckley et al. (2016) which separates these two species. This article does not provide details on the modern reference material sampled, so it is unclear whether more than one individual per species has been sampled. Based on the mention in their acknowledgements that material was sampled from the Sheffield Reference collection and the reference to these materials with only one reference ID in supplementary table S3 from Harvey et al. (2019), it appears that only one reference specimen was sampled for each species. Coupled with the fact that they only performed LC-MS/MS on *R. norvegicus* and the two available genome assemblies for *Rattus rattus* and the annotated COL1a2 sequence for this species was not available until 2020, the marker for *R. rattus* was based solely on a peptide mass fingerprint from a single individual.

Besides listing the same markers based on the original data in Harvey et al. (2019b) and the COL1a2 sequence for *R. norvegicus* from Welker et al. (2016) a thorough interrogation of the validity of these markers has not been published, despite the later use of these species specific markers (e.g. in Guiry et al. (2024)). With that said, we are not arguing that this marker is incorrect, and in fact, we looked up the suggested peptide sequence for *R. rattus* and it matches the sequence predicted for the two now published genome assemblies. We would like to highlight, however, that this is an issue that exists within the literature.

The original marker for *Camelus bactrianus* and *Camelus dromedarius* separation was similarly derived from a single peptide fingerprint of *Camelus bactrianus* (Rybczynski et al., 2013). This was then cited in a paper as a marker to identify 20 samples as hybrid camel (Marković et al., 2021), without additional sampling of reference material, and despite the fact the peptide sequences presented in Rybczynski et al. (2013) differed from those presented in Marković et al. (2021). While hybrid camels are a well known phenomenon, a greater weight needs to be placed on the possibility of intraspecies variation for these types of studies going forward.

Finally, while many studies today attempt to sample multiple individuals per species/taxonomic group, there are many justifiable reasons why this is not always possible for every taxonomic group of interest. Yet, it is important to make it clear when biomarkers are presented how many individuals were used to assess a given marker. Janzen et al. (2021), for example, sampled three individuals where possible, and carefully described which markers can and can't be used for MALDI. However, they make no distinction between markers identified from three individuals and those from only one, as is the case for every species in Antilopinae. We agree that potential markers should of course be published when only one species is available, but we hope that it will become standard to publish tables of biomarkers which provide more data upfront, such as the number of species in which a marker has been observed.

- 5) The statement that the level of taxonomic identification has not been systematically examined is also not necessarily supported and discounts a decade of re-analysis of species included in the ZooMS reference library. The benchmark standard of creating ZooMS reference libraries is to include multiple individuals to avoid potential SAPs (see for instance Janzen et al., 2021 where three individuals per species were used to create reference**

databases). In instances where variation in the protein of interest is detected within individual specimens of the same species, these peptides are not used in ZooMS identifications. Since the mass spectrometers used for ZooMS analysis are not capable of the resolution achieved with LC-MSMS, the peptides used for taxonomic identification need to be observed reliably. SAPs are therefore excluded so that the results can be considered reliable without the use of LC-MSMS.

REPOSNE: As we mention in a previous comment, we have revised the text to clarify it is *intraspecies* diversity that has not been systematically examined. In addition, we agree that Janzen et al. (2021) is a good example, but argue that sampling multiple individuals is not yet the standard adopted in the field, nor has the reasoning been made explicit. We believe that in addition to what reviewer 2 describes, researchers need to publish the number of individuals within the same tables where new biomarkers are presented. The reviewer states that **“In instances where variation in the protein of interest is detected within individual specimens of the same species, these peptides are not used in ZooMS identifications.”** We don’t argue that this is not occurring or unreasonable, but we argue that when it does occur, it should be investigated further or made explicit in the publication (as in Madupe et al. 2023; Kubat et al. 2023; Tsutaya et al. 2025).

We have revised the discussion section (line 357) to more explicitly reference previous researchers who have sought to analyse three or more individuals per species. “A general trend is developing in the field that markers should be confirmed using both genomic and proteomic data, and that samples from modern reference materials should aim for at least three individuals for a given taxon (Dierickx et al., 2022; Janzen et al., 2021; Peters et al., 2021)”

6) As you also identify in your own research in the case of COL1, the proteins generally used for ZooMS identification show higher levels of conservation overall as SAPs are observed far less frequently observed than in other proteins. If your research had demonstrated that SAPs were rampant in COL1 then this would be an important call-to-arms, but 5 individual SAPs were identified across dozens of specimens. A single SAP within COL1 is also unlikely to completely change the taxonomic identification of a specimen as ZooMS identifications are based on a series of markers and peptides.

RESPONSE: While we agree with the general reasoning of the reviewer, many identifications do come down to a single SAP, given that collagen shows higher levels of conservation. Again, it is important to assess inter- vs intra-taxon variability.

We have clarified the text to highlight that we observed intraspecies SAPs in 11-18% of the species where multiple individuals are available for analysis (see response 4 above). We currently don’t know how frequently it may occur for other animals, but given that intraspecies variability in our dataset increased with interspecies and intergenus variability, intraspecies variability is likely more common in fish, amphibians, and small rodents, where greater interspecies variability has been observed. We believe that this observation does justify a call-to-arms.

- 6) **Line 52 - As ZooMS has grown, the same species have been re-studied multiple times. For instance the original database (Buckley et al., 2009) was republished and updated in Welker et al., 2016 alongside collagen sequences. Since then, many of those species have had multiple individuals re-analysed to try and identify new peptides for greater levels of identifications. The COL1 sequences of sheep for instance have been studied as part of Buckley et al., 2009, Welker et al., 2016, Janzen et al., 2021 and in: Coutu, A.N., Taurozzi, A.J., Mackie, M., Jensen, T.Z.T., Collins, M.J., Sealy, J., 2021. Palaeoproteomics confirm earliest domesticated sheep in southern Africa ca. 2000 BP. Sci. Rep. 11, 6631.**

RESPONSE: While it is true that sheep markers have been replicated many times, most of the 75 available genome assemblies for sheep and 45 for goat were made available after Dec 2021. These include many different breeds and as far as we are aware, the collagen sequences from these genomes have never been examined for ZooMS applications. Given that researchers frequently argue that certain taxa have been morphologically misidentified based on ZooMS identifications (Seabrook et al. 2025 and Jeanjean et al 2023 are two recent examples), we argue it would be beneficial to confirm that genomic data support the single biomarker for sheep/goat discrimination.

8) **Line 55 - ZooMS does not rely on groupings like genera, it uses this language to simplify discussion of the method (by saying things like genus or family level identifications) but the taxonomic groupings it can achieve frequently do not follow genus groupings. In instances in which fauna can be differentiated from other members of the same genera, this is discussed in the literature. Brown et al., 2021 for instance re-classifies their groupings as “ZooMS taxons” as their identifications do not follow zooarchaeological categories previously identified at the site of interest. The oversimplification in my view, is in the manner in which ZooMS is discussed in your introductory paragraphs and not in the way that other researchers in the field are applying the method - particularly those working on improving reference databases. See for example:**

- Jensen, T.Z.T., Sjöström, A., Fischer, A., Rosengren, E., Lanigan, L.T., Bennike, O., Richter, K.K., Gron, K.J., Mackie, M., Mortensen, M.F., Sørensen, L., Chivall, D., Iversen, K.H., Taurozzi, A.J., Olsen, J., Schroeder, H., Milner, N., Sørensen, M., Collins, M.J., 2020. An integrated analysis of Maglemose bone points reframes the Early Mesolithic of Southern Scandinavia. *Sci. Rep.* 10, 17244. (Which includes a lengthy supplementary on the complexity of identifying peptide markers suitable for ZooMS analysis)
- Janzen, A., Richter, K.K., Mwebi, O., Brown, S., Onduso, V., Gatwiri, F., Ndiema, E., Katongo, M., Goldstein, S.T., Douka, K., Boivin, N., 2021. Distinguishing African bovids using Zooarchaeology by Mass Spectrometry (ZooMS): New peptide markers and insights into Iron Age economies in Zambia. *PLoS One* 16, e0251061. (Which discusses several markers commonly used in ZooMS analysis that should be considered unreliable and highlights the complexities of identifying peptide markers suitable for ZooMS analysis)
- Dierickx, K., Presslee, S., Hagan, R., Oueslati, T., Harland, J., Hendy, J., Orton, D., Alexander, M., Harvey, V.L., 2022. Peptide mass fingerprinting of preserved collagen in archaeological fish bones for the identification of flatfish in European waters. *R Soc Open Sci* 9, 220149.

- Korzow Richter, K., McGrath, K., Masson-MacLean, E., Hickinbotham, S., Tedder, A., Britton, K., Bottomley, Z., Dobney, K., Hulme-Beaman, A., Zona, M., Fischer, R., Collins, M.J., Speller, C.F., 2020. What's the catch? Archaeological application of rapid collagen-based species identification for Pacific Salmon. *J. Archaeol. Sci.* 116, 105116.
- Peters, C., Richter, K.K., Manne, T., Dortch, J., Paterson, A., Travouillon, K., Louys, J., Price, G.J., Petraglia, M., Crowther, A., Boivin, N., 2021. Species identification of Australian marsupials using collagen fingerprinting. *R Soc Open Sci* 8, 211229.

RESPONSE: We thank reviewer 2 for their thorough comments and concede that our introduction oversimplified the discussion of ZooMS. We believe we have sufficiently addressed this point in the preceding comments and the revisions previously outlined.

9) Line 286 and Supplementary: I think it is important in the main text to say that you were able to refine the identifications made in the Codlin paper through the use of LC-MS/MS and not MALDI-TOF. In the supplementary, you describe not being able to use the SAPs identified in this manuscript to refine the identifications from the Codlin paper through peptides which are visible in MALDI-TOF spectra but this is not as clear in the main text. Most people familiar with the field will accept that LC-MS/MS data will generally return a higher degree of taxonomic resolution than MALDI-TOF MS.

RESPONSE: We state in the supplementary note 3 *"In three out of four of these duck groups, taxonomic identification can be made based on MALDI-TOF MS data alone, while group 2 requires multiple other biomarkers best observed with LC-MS/MS data."* When Codlin et al. (2022) was published, there were not enough Anatidae collagen sequences available to identify these ducks beyond family level. Among the theoretical peptide masses from the sequences in dataset 2, there is only one peptide with a mass for 2777.4 *m/z* that overlaps with the markers identified. That peptide has a low probability of being produced by peptide digestion (Gasteiger et al., 2005; Keil, 2012), and indeed it has not been observed in any MALDI spectra where the sequence is present, which includes all of the duck taxa from Teotihuacan. For group 2, there is an additional SAP that may help further discriminate among the taxa listed in Supplementary Table 12. Since the duck species at Teotihuacan are now well represented in the database, these markers are suitable for MALDI identification of the species in this region listed in Supplementary Table 12.

We have updated the supplementary material (line 146) to clarify this:

"While there is a theoretical peptide found in many COL1a2 sequences with a predicted *m/z* = 2777.4, this peptide requires a missed trypsin cleavage and the mass is not observed in the spectra of other species that share this peptide sequence, including all Anatidae discussed here. The LC-MS/MS data for sample MC148 and non-Anatidae taxa from Codlin et al. (2022) with spectra presenting a peak at *m/z* 2777.3 confirms that the mass derives from the peptide sequence predicted by the COL1a2 sequences for *Mareca*."

7) Line 311: I think it is important to note here that such standards have been called for (as the authors indicate by referencing Richter's recent review) in previous literature and are already being followed by many people working in

the field. Although I appreciate such a direct list of standards the field should adopt as a whole

RESPONSE: We thank you for your suggestion and have added reference to previous literature that our standards build upon. We have revised the discussion (line 351) “This means there is an urgent need to expand existing standards for palaeoproteomic analysis (Brown et al., 2021; Hendy et al., 2018) and improve existing standards for identifying taxon-specific “marker” SAPs for both collagen (as predicted by Richter et al(2022)) and non collagenous proteins.” We also provide more explicit reference to other recent papers where multiple individuals have been explicitly sampled (see response to comment 5)

Reviewer #2 (Remarks on code availability):

All of the data is made available. A really wonderful example of data sharing!

Reviewer #3 (Remarks to the Author):

This is a great paper that highlights the need for more in depth assessment of proteomic data acquired from archaeological material, when making taxonomic identifications. The paper is well written, easy to follow and all the data has been critically examined with well-reasoned and clear results. The authors make several recommendations which are well justified and should be embraced (where possible) by the field of paleo proteomics.

All the data has been clearly reported and is accessible

I only have two minor comments:

- 1) Line 283 – consider having a subheading for the arch case study to make this clearer in the discussion**

RESPONSE: While we agree with the reviewer, the guidelines for this journal do not allow subheadings in the discussion. To help signpost this transition, we have moved the last sentence of the preceding paragraph to the beginning of the first case study paragraph.

- 2) Line 341 – typo for overcome**

RESPONSE: Thank you, we have fixed this

Reviewer #3 (Remarks on code availability):

The access to the dataset was restricted until publication. Only the metadata was available to view

RESPONSE: We apologise for this issue and are not sure of the cause. We have confirmed the link now works as expected to provide complete access to the data.

Reviewer #4 (Remarks to the Author):

Mz primary expertise is in the field of proteomics and analytical sciences, and as such my review does not address the construction of phylogenetic trees in great detail. From my point of view, the manuscript is scientifically sound, with a well-detailed methodology and rigorous data-analysis, grammatically well-written, and addresses some very relevant questions in the field of paleoproteomics. Even though it has been argued that proteomics is a tool for identifying the protein and not necessarily the species, paleoproteomic studies are increasingly used to address species-specific archaeological questions, and as such this manuscript is relevant in trying to analyse some of the challenges for using proteomics for such purpose.

However, I do have a few questions regarding the manuscript.

- 1) Very broadly, the manuscript (including the abstract), talks about the widespread evidence for SAPs occurring within species. However, from the data it appears that the vast majority of the SAPs observed are concentrated on eggshell proteins. As such, can the authors please specify this in the abstract and the introduction while discussing the presence of SAPs in general?**

RESPONSE: we agree with the reviewer and have updated the abstract:

“While palaeoproteomics (the study of ancient proteins by mass spectrometry) conventionally assumes that protein sequences vary only at the species level or above, our research demonstrates widespread evidence for single amino acid polymorphisms (SAPs) occurring within-species, particularly within the main avian eggshell proteins.” (line 20)

However, we have decided not to update the introduction as this would not fit into the current flow of the section.

- 2) To me, the two main findings of the manuscript can occasionally appear somewhat paradoxical and contradictory. Whereas the discussion about intraspecies diversity warns about the challenges regarding the use of SAPs for taxonomic distribution, the archaeological case studies mentioned introduces new reference sequence and uses them for taxonomic identification to genus and species level based on the presence of SAPs. Although some of this is discussed in the Supplementary text (page 7 and 8), it is not sufficiently clear to me how the authors distinguish between the SAPs they use for taxonomic identification of the various Anatidae species with SAPs possibly arising as a result of the reported intra/interspecies variation. Can the authors please clarify this, and perhaps introduce a separate paragraph in the discussion section addressing this point in sufficient detail?**

RESPONSE: These are indeed two paradoxical outcomes of our work, and stem from having a large protein database (or rather a larger protein database that is commonly available). Our goal is to highlight the challenges of intraspecies variability so that we may address them sufficiently to be able to make confident taxonomic identifications. We have added some additional text to ensure our reasoning for taxonomic identifications are clear.

In main text where we make the identification for Tlajinga we have added “...used dataset 2 to refine identifications, relying on SAPs that are observed consistently across multiple individuals from the taxonomic groups concerned.” (line 319)

And the Shubayqa case study in main text now reads (line 330):

“Using dataset 2, we reassessed the four published LC-MS/MS analyses available for these samples and identified multiple consistent SAPs across proteins. These allow us to attribute the previously unidentified duck to either shelduck (*Tadorna* sp.) or Egyptian goose (*Alopochen aegyptiaca*). *Alopochen aegyptiaca* is less likely because it is not known to occur naturally in Jordan (Andrews, 1995). This taxon cannot be ruled out on a molecular basis because its protein sequences are not available in the database, as only a single genome, with low coverage across some eggshell genes, was available. . Meanwhile, *Tadorna* is known to occasionally breed in the region today (Andrews, 1995) and the peptides detected in the archaeological sample match the available protein reference sequences for this taxon. (Supplementary Note 3). The *Anser* or *Branta* goose specimen is now securely identified as *Anser* sp., based on SAPs that consistently separate the two taxa in XCA1, and XCA2, OC116 and BPIfoldB4 peptides, although cannot be identified to species given the observed intraspecies variability in *Anser* OC116 sequences. The swan, in contrast, matched uniquely to protein sequences found in mute swan (*Cygnus olor*), where multiple SAPs across XCA2 and OC116 distinguish this species from all other individuals belonging to this genus...

To the methods section we have added (line 592):

“Identifications were made by examining the range of variation for the SAPs observed in the archaeological samples, giving weight to SAPs that are consistent across a range of individuals or species. In the case of *Cygnus olor*, identification was made despite the presence of only one individual in the database. This is due to the fact there are multiple SAPs shared between the archaeological specimen and the phased genomic data available for *C. olor*. More specifically, the identification was made based on matches to SAPs that are either not found in other species of *Cygnus*, or that are shared between *C. olor*, *C. melancoryphus* or *C. atratus* - the latter two being birds native to South America and Australia respectively.”

In our revisions of the supplementary note 3 we focus mainly on the Shubayqa example which was less concretely explained based on the higher number of SAPs involved. These sections now include information about peptide spectrum matches unique to specific proteins.

(line 193) “With the larger dataset, we can confirm that the unidentified duck with a distinctive marker at m/z 2461.2 belonged to a species not present in the original database. *Tadorna* ($n^i=4$, $n^{sp}=2$), *Alopochen* ($n^i=1$) and *Plectropterus* ($n^i=1$) all have the SAP in the XCA2 sequence which results in the marker at m/z 2461.2. Therefore they cannot be distinguished by MALDI-MS. However, there are 36 SAPs in OC116 and 18 SAPs in BPIfoldB4 that separate the single *Plectropterus* genome from the consensus sequence of *Tadorna* ($n^i=5$, $n^{sp}=3$). For each of these two proteins, we found that 7-50% more peptides matched to *Tadorna* than to *Plectropterus* (Supplementary Tables 26 and 29), hence we are confident in ruling out *Plectropterus*. Only one *Alopochen* genome was available, and the lower quality of this genome means that fewer proteins were successfully annotated for this species, making it difficult to rule out based on protein sequence information. ...”

(line 212) “The LC-MS/MS data for the sample previously identified at *Cygnus* sp. matched most closely to protein sequences found in *Cygnus olor* (Supplementary Tables 32-4) .

Cygnus olor diverged from other swans found in the northern hemisphere, *C. cygnus*, *C. columbianus* and *C. buccinator* around 7.5 mya (according to Sun et al.(2017)), which is reflected as SAP differences between these species in every protein in our study with the exception of Ovalbumin (see Supplementary Figures 1-13). While only one *C. olor* individual (two sequences) is present in our database, there are seven individuals in total for the *Cygnus* genus ($n^{sp}=6$). The PEAKS data for the archaeological sample identified 72 peptide matches to *C. olor*, compared to 56 to *C. cygnus* for OC116, and 58 peptide matches to *C. olor* compared to 28 to *C. buccinator*, *C. cygnus* and *C. columbianus*. Moreover, while all three *C. olor* SAPs identified in the archaeological XCA2 sequence are also found in *C. atratus* ($n^i=1$), this species is native to Australia. Overall, this suggests that many of the SAPs identified in the archaeological sample are not the result of intraspecies variation, supporting the identification as *C. olor*...

(line 229) Multiple consistent SAPs in both XCA1 and XCA2 proteins support the separation of *Anser* and *Branta* geese (Supplementary Figure 16). The goose specimen from Shubayqa displays amino acid sequences found in the *Anser* genus for these proteins, including 35 peptides mapping only to *Anser* XCA1 peptides and 23 mapping only to *Anser* XCA2 according to the PEAKS algorithm (Supplementary Table 35). While the archaeological sample has 146 peptide matches to the OC116 sequence from *A. cygnoides* (GCA_013030995.1) compared to the next highest hit (136) to *A. indicus* (GCA_006229135.1), the intraspecies variation observed in OC116 for both species prevents us from more precisely identifying this specimen until interspecies variability observed in OC116 can be distinguished from intraspecies.

- 3) Can the authors please briefly discuss that the mass change corresponding to how many (if any) of the SAPs they observe can coincide with mass changes due to various observed PTMs in the mass spectra? (For e.g. the mass change due to an Q → R substitution can coincide with the mass change due to formylation, or the mass change due to P → Q substitution can coincide with a double oxidation or the oxidation of methionine to sulphone, etc.). Although not directly related to the manuscript, I believe that this will be helpful when it comes to interpreting tandem mass spectrometric data for species identification purposes.**

RESPONSE: We agree that this is a very pressing concern, which is often underestimated. We do not think we can conduct this assessment in a systematic manner in this study, but we are aware of it and tend to visually assess the frequency and likelihood of PTMs when evaluating our data. We refer Reviewer 4 to our supplementary tables 16, 19, 22, 25, 28, 31, 34, 37 where we provide the complete list of peptide spectrum matches (PSMs) for archaeological samples which can be filtered by PTM to assess their impact on peptide matches. A specific example for the Q/R SAP that distinguishes *Anser* and *Branta* can be observed in Supp. Figure 16 (peptide R(Q)ELSDCTPGWVPV). Only 6 PSMs match to the *Branta* QELSDCTPGWVPV sequence and all are modified with either Acetylation, Carbamylation or Glycidamide adduct (Supp. Table 37). In contrast, there are 210 PSMs to the *Anser* sequence, 93 of which contain no PTM besides Carbamidomethylation.

4) Line 82. Was the value of 70% chosen arbitrarily, or based on previous work/ existing convention?

This was not based on convention but was chosen to be a reasonable cut-off point for inclusion for analysis. We believe this cut-off maximises the number of species that are taken into account while minimising the issue that short sequences can lead to underestimate or overestimate similarities between species

5) Line 124. I think this line is very important, particularly given that single amino acid polymorphisms of collagen have been previously used to argue for taxonomic identification of ancient human remains (Chen et al. 2019). Although possibly outside the scope of this paper, it will be very interesting to see if this trend of SAPs hold on mammalian/primate species, given that the majority of the SAPs observed here seem to be associated with proteins involved in egg-laying and, more generally, reproduction.

RESPONSE: We thank the reviewer for this comment. In fact, with the publication of a new proteomic study on Denisovans (Tsutaya et al. 2025: Table 1) there is additional evidence for intraspecies variation and heterozygosity in hominin collagen sequences. We have updated our introduction (response 3, reviewer 2) and discussion (response 4, reviewer 2) to reflect the contribution of this new paper and linked it to ZooMS analysis of taxa beyond mammals and birds.

We have also added to our discussion a section talking about the MEPE homologue in mammals which is not related to reproduction. (see also response below)

“Moreover, while most of the intraspecies variation we observed occurs in eggshell proteins associated with reproduction, many of these sequences have homologues in other animals, including ovocalyxin-32 and OC116 (Le Roy et al., 2021). Avian ovocleidin 116 is homologous to the matrix extracellular phosphoglycoprotein (MEPE) in mammals that plays a role in biomineralisation of bone (Bardet et al., 2010). As of 18/04/2025, there are at least 721 observed variants for Human MEPE on UniProt (UniProt Consortium, 2025) indicating that this variation is not related specifically to avian reproduction.” (line 271)

6) Line 177. From supplementary table 4, it appears that OC116 has the highest number of SAPs based on mean pairwise by far among (6,4 vs 2,9) all the proteins considered for Anatidae. Can the authors please perform the annotation for the seven additional species described here using a protein other than OC116, for e.g. ovotransferrin or albumin)?

RESPONSE: We agree that this is an important issue, and we have performed additional annotations with Ovocalyxin-32. We chose this protein over ovotransferrin, as there were issues finding suitable ovotransferrin proteins for annotations in Anatidae, and over albumin, as this was found to have very low variability in our study. Additionally, we thought it would be a good complement to OC116 as both of these proteins also have well-studied mammalian homologues. Of the seven non Anatidae species we annotated for Ovocalyxin-32, only six species provided more than one sequence for assessing intraspecies variability, and SAPs were observed in three of these (after trimming the first 49 amino acids of the Ovocalyxin-32 sequence, which showed too much variation to establish a

consensus sequence for the protein). We have updated the text to reflect the additional data, as well as supplementary table 8 and the methods to reflect this inclusion.

To the results we have added (line 194):

“We observed intraspecies variability in Ovocalyxin-32 in *Grus americana* (whooping crane), *Chlamydotis macqueenii* and *Lycocorax pyrrhopterus obiensis*.”

- 7) **Line 184. Given that collagen is the protein most often used for taxonomic identification and phylogenetic studies based on proteomics, this section should be emphasized and discussed in greater detail. To me, this section reads that the presence of SAPs in collagen is poorly supported, and if that is correct, the effect of SAPs on collagen-based taxonomic identification will be minimal, and the manuscript needs to reflect that. In my opinion, the discussion section needs to have an additional paragraph discussing that importance of the protein (and indirectly, substrate) used for proteomics-based taxonomic identification- from my understanding, the proteins involved in eggshells and/or other reproductive traits seem to be substantially more affected by SAPs as compared to collagen. If that understanding is correct, some of the primary findings of the paper, although extremely relevant to eggshell-based proteomics, are perhaps less relevant for proteins extracted from other substrates like bones, and this needs to be explicitly mentioned in the manuscript.**
- 8) **Line 308. Please can the authors clarify this statement, given that the presence of SAPs in collagen are much lower as compared to eggshell proteins (acc to Supp. Table 4) and previously in line 184 they state at least at least some of the SAPs observed in collagen is poorly supported?**

RESPONSE to comments 8&9 (interrelated): We thank the reviewer for this comment. We agree that applications employing collagen currently dominate the literature, and so have added a section discussing the implications of our findings to other taxa, especially those animals where collagen seems to be more variable at the species level, like fish and amphibians. However, we do think that although more variation is found in eggshells, these findings are relevant and should be taken into careful consideration for all substrates. For example, we highlight that OC116 is a homologue to MEPE, a protein found in human bone, which has hundreds of known variants (see response to comment 5). Furthermore, we point out another of our key findings, i.e. the differing topology in COL1a1 and COL1a2 phylogenetic trees, and how these differ to the other proteins (Figure 2). This calls into question how well collagen can be used to create phylogenetic trees for species placement.

The changes made to the text to reflect these ideas have been provided in our responses to previous comments.

Finally, the finding that the SAP among the two *Anser indicus* genomes was poorly supported is due to the error occurring at the genome assembly level, rather than due to any major issue in the annotation process. We do not think that this outweighs the fact that intraspecies SAPs in collagen are not rare when considering the number of species in which they were observed, i.e. ~18% for COL1a1 and ~11% for COL1a2 (as indicated by Supp.

Table 4 and Figure 3). As detailed in a previous response to reviewer 2, we have updated the text to reflect this.

Reviewer #4 (Remarks on code availability):

I have reviewed the protein analysis of the code in detail, but the construction of the phylogenetic trees are outside my immediate area of expertise.

References cited in response

- Andrews, I. J. (1995). *The birds of the Hashemite Kingdom of Jordan*. I.J. Andrews.
- Bardet, C., Vincent, C., Lajarille, M.-C., Jaffredo, T., & Sire, J.-Y. (2010). OC-116, the chicken ortholog of mammalian MEPE found in eggshell, is also expressed in bone cells. *Journal of Experimental Zoology. Part B, Molecular and Developmental Evolution*, 314B(8), 653–662. <https://doi.org/10.1002/jez.b.21366>
- Brown, S., Douka, K., Collins, M. J., & Richter, K. K. (2021). On the standardization of ZooMS nomenclature. *Journal of Proteomics*, 235, 104041. <https://doi.org/10.1016/j.jprot.2020.104041>
- Buckley, M. (2018). Zooarchaeology by Mass Spectrometry (ZooMS) Collagen Fingerprinting for the Species Identification of Archaeological Bone Fragments. In C. M. Giovas & M. J. LeFebvre (Eds.), *Zooarchaeology in Practice* (pp. 227–247). Springer International Publishing. https://doi.org/10.1007/978-3-319-64763-0_12
- Buckley, M., & Cheylan, M. (2020). Collagen fingerprinting for the species identification of archaeological amphibian remains. *Boreas*, 49(4), 709–717. <https://doi.org/10.1111/bor.12443>
- Buckley, M., Collins, M., Thomas-Oates, J., & Wilson, J. C. (2009). Species identification by analysis of bone collagen using matrix-assisted laser desorption/ionisation time-of-flight mass spectrometry. *Rapid Communications in Mass Spectrometry*, 23(23), 3843–3854. <https://doi.org/10.1002/rcm.4316>
- Buckley, M., Gu, M., Shameer, S., Patel, S., & Chamberlain, A. T. (2016). High-throughput collagen fingerprinting of intact microfaunal remains; a low-cost method for distinguishing between murine rodent bones. *Rapid Communications in Mass Spectrometry: RCM*, 30(7), 805–812. <https://doi.org/10.1002/rcm.7483>
- Buckley, M., & Kansa, S. W. (2011). Collagen fingerprinting of archaeological bone and teeth remains from Domuztepe, South Eastern Turkey. *Archaeological and Anthropological Sciences*, 3(3), 271–280. <https://doi.org/10.1007/s12520-011-0066-z>
- Codlin, M. C., Douka, K., & Richter, K. K. (2022). An application of zooms to identify

- archaeological avian fauna from Teotihuacan, Mexico. *Journal of Archaeological Science*, 148, 105692. <https://doi.org/10.1016/j.jas.2022.105692>
- Dierickx, K., Presslee, S., Hagan, R., Oueslati, T., Harland, J., Hendy, J., Orton, D., Alexander, M., & Harvey, V. L. (2022). Peptide mass fingerprinting of preserved collagen in archaeological fish bones for the identification of flatfish in European waters. *Royal Society Open Science*, 9(7), 220149. <https://doi.org/10.1098/rsos.220149>
- Gasteiger, E., Hoogland, C., Gattiker, A., Duvaud, S., 'everine, Wilkins, M. R., Appel, R. D., & Bairoch, A. (2005). Protein identification and analysis tools on the ExPASy server. In *The Proteomics Protocols Handbook* (pp. 571–607). Humana Press. <https://doi.org/10.1385/1-59259-890-0:571>
- Guiry, E., Kennedy, R., Orton, D., Armitage, P., Bratten, J., Dagneau, C., Dawdy, S., deFrance, S., Gaulton, B., Givens, D., Hall, O., Laberge, A., Lavin, M., Miller, H., Minkoff, M. F., Niculescu, T., Noël, S., Pavao-Zuckerman, B., Stricker, L., ... Buckley, M. (2024). The ratting of North America: A 350-year retrospective on *Rattus* species compositions and competition. *Science Advances*, 10(14), eadm6755. <https://doi.org/10.1126/sciadv.adm6755>
- Harvey, V. L., Daugnora, L., & Buckley, M. (2018). Species identification of ancient Lithuanian fish remains using collagen fingerprinting. *Journal of Archaeological Science*, 98, 102–111. <https://doi.org/10.1016/j.jas.2018.07.006>
- Harvey, V. L., Egerton, V. M., Chamberlain, A. T., Manning, P. L., Sellers, W. I., & Buckley, M. (2019). Interpreting the historical terrestrial vertebrate biodiversity of Cayman Brac (Greater Antilles, Caribbean) through collagen fingerprinting. *The Holocene*, 29(4), 531–542. <https://doi.org/10.1177/0959683618824793>
- Harvey, V. L., LeFebvre, M. J., deFrance, S. D., Toftgaard, C., Drosou, K., Kitchener, A. C., & Buckley, M. (2019). Preserved collagen reveals species identity in archaeological marine turtle bones from Caribbean and Florida sites. *Royal Society Open Science*, 6(10), 191137. <https://doi.org/10.1098/rsos.191137>
- Hendy, J., Welker, F., Demarchi, B., Speller, C., Warinner, C., & Collins, M. J. (2018). A guide

to ancient protein studies. *Nature Ecology & Evolution*, 2(5), 791–799.

<https://doi.org/10.1038/s41559-018-0510-x>

Janzen, A., Richter, K. K., Mwebi, O., Brown, S., Onduso, V., Gatwiri, F., Ndiema, E., Katongo, M., Goldstein, S. T., Douka, K., & Boivin, N. (2021). Distinguishing African bovids using Zooarchaeology by Mass Spectrometry (ZooMS): New peptide markers and insights into Iron Age economies in Zambia. *PLoS One*, 16(5), e0251061.

<https://doi.org/10.1371/journal.pone.0251061>

Jensen, T. Z. T., Sjöström, A., Fischer, A., Rosengren, E., Lanigan, L. T., Bennike, O., Richter, K. K., Gron, K. J., Mackie, M., Mortensen, M. F., Sørensen, L., Chivall, D., Iversen, K. H., Taurozzi, A. J., Olsen, J., Schroeder, H., Milner, N., Sørensen, M., & Collins, M. J. (2020). An integrated analysis of Maglemose bone points reframes the Early Mesolithic of Southern Scandinavia. *Scientific Reports*, 10(1), 17244.

<https://doi.org/10.1038/s41598-020-74258-8>

Keil, B. (2012). *Specificity of Proteolysis*. Springer Science & Business Media.

<https://play.google.com/store/books/details?id=NAH9CAAAQBAJ>

Korzow Richter, K., McGrath, K., Masson-MacLean, E., Hickinbotham, S., Tedder, A., Britton, K., Bottomley, Z., Dobney, K., Hulme-Beaman, A., Zona, M., Fischer, R., Collins, M. J., & Speller, C. F. (2020). What's the catch? Archaeological application of rapid collagen-based species identification for Pacific Salmon. *Journal of Archaeological Science*, 116, 105116. <https://doi.org/10.1016/j.jas.2020.105116>

Kubat, J., Paterson, R., Patramanis, I., Barker, G., Demeter, F., Filoux, A., Kullmer, O., Mackie, M., Marques-Bonet, T., Huong, N. T. M., Tuan, N. A., Pheng, S., Rippengal, J., Schrenk, F., Souksavatdy, V., Tshen, L. T., Wattanapituksakul, A., Wang, W., Zanolli, C., ... Bacon, A.-M. (2023). Geometric morphometrics and paleoproteomics enlighten the paleodiversity of Pongo. *PLOS ONE*, 18(12), e0291308.

<https://doi.org/10.1371/journal.pone.0291308>

Le Roy, N., Stapane, L., Gautron, J., & Hincke, M. T. (2021). Evolution of the Avian Eggshell Biom mineralization Protein Toolkit – New Insights From Multi-Omics. *Frontiers in*

Genetics, 12. <https://doi.org/10.3389/fgene.2021.672433>

- Madupe, P. P., Koenig, C., Patramanis, I., R  ther, P. L., Hlazo, N., Mackie, M., Tawane, M., Krueger, J., Taurozzi, A. J., Troch  , G., Kibii, J., Pickering, R., Dickinson, M., Sahle, Y., Kgotleng, D., Musiba, C., Manthi, F., Bell, L., DuPlessis, M., ... Cappellini, E. (2023). Enamel proteins reveal biological sex and genetic variability within southern African Paranthropus. In *bioRxiv*. bioRxiv. <https://doi.org/10.1101/2023.07.03.547326>
- Markovi  , N., Ivani  evi  , V., Baron, H., Lawless, C., & Buckley, M. (2021). The last caravans in antiquity: Camel remains from Cari  in Grad (Justiniana Prima). *Journal of Archaeological Science, Reports*, 38(103038), 103038. <https://doi.org/10.1016/j.jasrep.2021.103038>
- Meinertzhagen, R. (1935). Ornithological results of a trip to Syria and adjacent countries 1933. *Ibis*, 13, 110–151.
- Peters, C., Richter, K. K., Manne, T., Dortch, J., Paterson, A., Travouillon, K., Louys, J., Price, G. J., Petraglia, M., Crowther, A., & Boivin, N. (2021). Species identification of Australian marsupials using collagen fingerprinting. *Royal Society Open Science*, 8(10), 211229. <https://doi.org/10.1098/rsos.211229>
- Richter, K. K., Codlin, M. C., Seabrook, M., & Warinner, C. (2022). A primer for ZooMS applications in archaeology. *Proceedings of the National Academy of Sciences of the United States of America*, 119(20), e2109323119. <https://doi.org/10.1073/pnas.2109323119>
- Rybczynski, N., Gosse, J. C., Harington, C. R., Wogelius, R. A., Hidy, A. J., & Buckley, M. (2013). Mid-Pliocene warm-period deposits in the High Arctic yield insight into camel evolution. *Nature Communications*, 4(1), 1550. <https://doi.org/10.1038/ncomms2516>
- Sun, Z., Pan, T., Hu, C., Sun, L., Ding, H., Wang, H., Zhang, C., Jin, H., Chang, Q., Kan, X., & Zhang, B. (2017). Rapid and recent diversification patterns in Anseriformes birds: Inferred from molecular phylogeny and diversification analyses. *PLoS One*, 12(9), e0184529. <https://doi.org/10.1371/journal.pone.0184529>
- Tsutaya, T., Sawafuji, R., Taurozzi, A. J., Fagern  s, Z., Patramanis, I., Troch  , G., Mackie,

- M., Gakuhari, T., Oota, H., Tsai, C.-H., Olsen, J. V., Kaifu, Y., Chang, C.-H., Cappellini, E., & Welker, F. (2025). A male Denisovan mandible from Pleistocene Taiwan. *Science (New York, N.Y.)*, 388(6743), 176–180. <https://doi.org/10.1126/science.ads3888>
- UniProt Consortium. (2025). UniProt: The universal protein knowledgebase in 2025. *Nucleic Acids Research*, 53(D1), D609–D617. <https://doi.org/10.1093/nar/gkae1010>
- Wallace, D. I. M. (1983). The breeding birds of the Azraq Oasis and its desert surround, Jordan, in the mid-1960's. *Sandgrouse*, 5, 1–18.
- Welker, F., Hajdinjak, M., Talamo, S., Jaouen, K., Dannemann, M., David, F., Julien, M., Meyer, M., Kelso, J., Barnes, I., Brace, S., Kamminga, P., Fischer, R., Kessler, B. M., Stewart, J. R., Pääbo, S., Collins, M. J., & Hublin, J.-J. (2016). Palaeoproteomic evidence identifies archaic hominins associated with the Châtelperronian at the Grotte du Renne. *Proceedings of the National Academy of Sciences of the United States of America*, 113(40), 11162–11167. <https://doi.org/10.1073/pnas.1605834113>
- Yeomans, L., Codlin, M. C., Mazzucato, C., Dal Bello, F., & Demarchi, B. (2024). Waterfowl eggshell refines palaeoenvironmental reconstruction and supports multi-species niche construction at the Pleistocene-Holocene transition in the Levant. *Journal of Archaeological Method and Theory*, 31(3), 1383–1429. <https://doi.org/10.1007/s10816-024-09641-0>